# Proximity-magnetized quantum spin Hall insulator: monolayer 1 T' WTe$_2$/Cr$_2$Ge$_2$Te$_6$

Junxue Li[1,2,8], Mina Rashetnia[1,8], Mark Lohmann [1], Jahyun Koo[3], Youming Xu[4], Xiao Zhang[5], Kenji Watanabe [6], Takashi Taniguchi [7], Shuang Jia [5], Xi Chen[4], Binghai Yan [3], Yong-Tao Cui [1] & Jing Shi [1] ✉

Van der Waals heterostructures offer great versatility to tailor unique interactions at the atomically flat interfaces between dissimilar layered materials and induce novel physical phenomena. By bringing monolayer 1 T' WTe$_2$, a two-dimensional quantum spin Hall insulator, and few-layer Cr$_2$Ge$_2$Te$_6$, an insulating ferromagnet, into close proximity in an heterostructure, we introduce a ferromagnetic order in the former via the interfacial exchange interaction. The ferromagnetism in WTe$_2$ manifests in the anomalous Nernst effect, anomalous Hall effect as well as anisotropic magnetoresistance effect. Using local electrodes, we identify separate transport contributions from the metallic edge and insulating bulk. When driven by an AC current, the second harmonic voltage responses closely resemble the anomalous Nernst responses to AC temperature gradient generated by nonlocal heater, which appear as nonreciprocal signals with respect to the induced magnetization orientation. Our results from different electrodes reveal spin-polarized edge states in the magnetized quantum spin Hall insulator.

Conventional ferromagnets are known to exhibit a series of magnetization-dependent transverse transport phenomena as linear responses to electric field or temperature gradient, many of which are intimately interrelated. In metals and semiconductors for example, the anomalous Hall effect (AHE) and anomalous Nernst effect (ANE) are connected by the Mott relation[1–4] through the anomalous Hall conductivity's energy derivative, and their respective unquantized coefficients are quantitatively determined by either intrinsic and/or extrinsic mechanisms. In contrast to well-studied ferromagnetic systems, little is known about these properties in two-dimensional topological systems with edge states. In quantum anomalous Hall insulators (QAHI), for example, the anomalous Hall conductance is quantized to $\frac{e^2}{h}$ due to the one-dimensional (1D) ballistic chiral edge transport, the hallmark of the quantum anomalous Hall effect[5,6]. According to the Mott relation, a

1D ballistic chiral edge is not expected to generate any ANE. Experimentally it is challenging to investigate the ANE in QAHIs well below 1 K. In ideal quantum spin Hall insulators (QSHI)[7–14], on the other hand, the two counter-propagating helical edge currents produce neither Hall nor Nernst signal due to time reversal symmetry (TRS), although each edge channel has the same but opposite quantized Hall conductance. To understand the edge current transport between the fully spin-polarized in QAHIs and spin unpolarized in QSHIs, here we introduce a ferromagnetic order in a QSHI to study the behaviors of partially spin-polarized edge current transport.

Among existing 2D QSHIs including HgTe/CdTe[10,11] and InAs/GaSb[14] quantum wells, 1 T' monolayer (ML)-WTe$_2$ has recently attracted much attention for its well-defined bulk and gapless edge band structure with a large QSHI gap[15–17], spatially resolved metallic edge

[1]Department of Physics and Astronomy, University of California, Riverside, CA 92521, USA. [2]Department of Physics, Southern University of Science and Technology, Shenzhen 518055, China. [3]Department of Condensed Matter Physics, Weizmann Institute of Science, Rehovot, Israel. [4]Department of Electrical and Computer Engineering, University of California, Riverside, CA 92521, USA. [5]International Center for Quantum Materials, School of Physics, Peking University, Beijing 100871, China. [6]Research Center for Functional Materials, National Institute for Materials Science, 1-1 Namiki, Tsukuba 305-0044, Japan. [7]International Center for Materials Nanoarchitectonics, National Institute for Materials Science, 1-1 Namiki, Tsukuba 305-0044, Japan. [8]These authors contributed equally: Junxue Li, Mina Rashetnia. ✉e-mail: jing.shi@ucr.edu

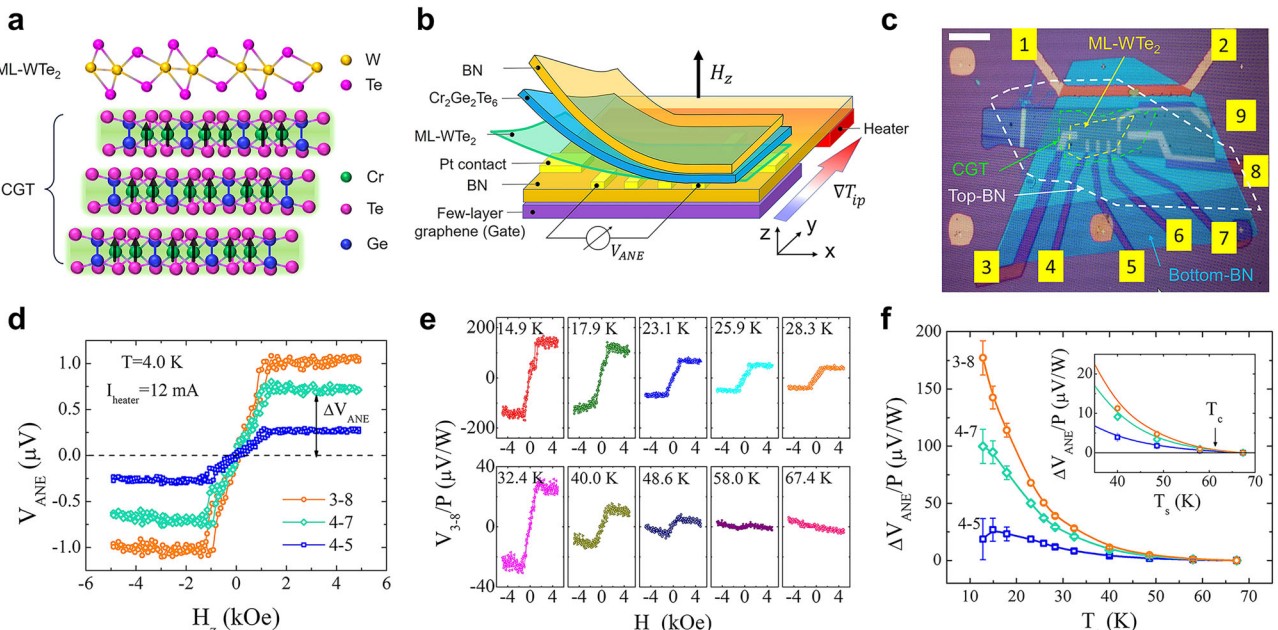

**Fig. 1 | Device structure and anomalous Nernst signals in monolayer 1 T' WTe₂/Cr₂Ge₂Te₆ heterostructure. a** Schematic of monolayer (ML) 1 T' WTe₂/Cr₂Ge₂-Te₆(CGT) vdW heterostructure. **b** Schematic of the ANE device structure. BN stands for hexagonal boron nitride. The nonlocal heater for generating in-plane temperature gradient $\nabla T_{ip}$ is underneath the bottom BN. Open circuit voltage $V_{ANE}$ due to ANE is measured in sweeping out-of-plane magnetic field $H_z$. **c** Device optical image before transfer of ML-WTe₂/CGT composite layer on pre-patterned Pt electrodes. The scale bar is 20 $\mu m$. The polygons with yellow, green and white dashed boundaries indicate the profiles of ML-WTe₂, CGT and top-BN, respectively. **d** Magnetic field dependence of 20-loop averaged ANE signal from electrode pairs 3–8, 4–7 and 4–5. The heating current in channel 1–2 is 12 mA and the system temperature is set to 4.0 K. **e** 20-loop averaged ANE signal from channel 3–8 at different sample temperatures ranging from 14.9 K to 67.4 K. The vertical axis is the ANE voltage normalized by the heating power $P$. The actual sample temperature $T_s$ is indicated in each panel which is calibrated with the temperature dependence of ML-WTe₂'s four-terminal resistance measured with a small current. **f** Heating-power-normalized ANE signals from channels 3–8, 4–7 and 4–5 vs. $T_s$. Inset shows the data between 40 K and 70 K, where the ANE signals in all three channels vanish around the Curie temperature $T_c = 61$ K of CGT.

states[18], possible exciton insulating state[19,20], and close-to-quantized conductance[21,22]. In general, ferromagnetic order can be introduced via doping or interfacial proximity coupling in heterostructures containing a ferromagnet. The latter approach has been demonstrated in several material systems including graphene[23] and topological insulators[24,25]. For ML-WTe₂, the heterostructure approach has a particular appeal because van der Waals (vdW) heterostructures such as CrI₃/WSe₂[26] and CrI₃/WTe₂[27] have proven very effective in creating proximity coupling due to the atomically flat interfaces. In CrI₃/WTe₂[27], edge current transport revealed fascinating nonlinear and non-reciprocal characteristics which were attributed to electron-magnon interaction between the metallic edge in WTe₂ and the magnetic CrI₃. It gave rise to interesting questions such as the magnetic state of the bulk and the nature of the nonreciprocity.

In this work, we fabricate high-quality vdW heterostructures comprised of ML-WTe₂ and few-layer vdW ferromagnet Cr₂Ge₂Te₆ (CGT) and probe edge and bulk transport responses to both AC temperature gradient and electric field. ANE and AHE as well as the anisotropic magnetoresistance (AMR) unequivocally affirm the proximity-induced ferromagnetism in the entire atomic layer of ML-WTe₂. At low temperatures, the two-component transport, i.e., the edge and bulk, can be cleanly disentangled. The unquantized AHE and definitive edge ANE responses are indicative of partially spin-polarized edge channel, distinguishing itself from the ideal 1D chiral or 1D helical edge channels.

## Results and discussion
We use a glove-box transfer/pickup technique to fabricate our heterostructure devices which is described in the Methods section and Supplementary Section 1. Figure 1a shows a schematic illustration of the ML-WTe₂/CGT vdW heterostructure. CGT is an insulating

ferromagnet (resistance well above ~10 GΩ in thin flakes) below its Curie temperature $T_c$ of 61 K with the magnetic anisotropy perpendicular to its atomic layers[28–31]. We expect a strong exchange interaction between ML-WTe₂ and CGT at the atomically flat interface, and consequently, for the former to acquire ferromagnetism via proximity coupling. We electrically probe the induced ferromagnetism in ML-WTe₂ by measuring the ANE, AMR and AHE responses. Since the transfer method is a low temperature process, it should not cause CGT to become conductive[31]. We also exclude the formation of a conductive surface layer of CGT by possible charge transfer from ML-WTe₂ because its resistance is found to increase after it is put in contact with CGT as will be discussed later. Due to the larger than four orders of magnitude difference in resistance between ML-WTe₂ and CGT, the transport responses should be solely from ML-WTe₂. The ANE device structure is shown in Fig. 1b. Figure 1c is the optical image of the device (D1) prior to transfer of ML-WTe₂/CGT composite layer by the pickup/transfer technique.

To measure ANE responses, we fabricate a heater for generating mainly a lateral temperature gradient $\nabla T$ perpendicular to both the heater and the voltage channel. While passing an AC current through the heater (via 1–2 electrode pair in Fig. 1c), we record the second harmonic voltage responses from electrode pairs 3–8, 4–7 and 4–5 (as shown in Fig. 1c) as an out-of-plane magnetic field $H_z$ is swept. Figure 1d summarizes the $H_z$-dependence of the voltage signals from the three electrode pairs at the nominal system temperature of 4.0 K. Hysteresis behaviors are observed in all three channels which resemble the anomalous Hall loop in CGT/Pt[31], except that here the slanted loops are nearly closed. We note that the shape of the hysteresis loop depends on CGT flake thickness and it can be completely collapsed above a certain thickness due to stronger dipolar interaction (see Fig. S-2 of ref. 31). The linear $H_z$-dependent background is often caused by the

ordinary Nernst effect from magnetic field but here it is much smaller than the magnitude of the total signal at saturation; therefore, the hysteresis loops cannot be produced by the stray field from CGT with saturation magnetization $M_s$ (<$4\pi M_s \sim 2$ kOe), which strongly indicates their acquired origin from CGT, i.e., the anomalous Nernst effect (ANE) arising from the spin-orbit coupling (SOC). The same argument holds for the SOC origin of the AHE in ferromagnets[3] simply because the ordinary Hall effect cannot account for the large magnitude of the Hall hysteresis loops if it was from a stray field-generated ordinary Hall signal. Additionally, in Supplementary Section 2, we present a discussion to exclude the spin Seebeck effect (SSE), the competing effect that can arise from unmagnetized WTe$_2$. The ANE voltage, i.e., $V_{ANE} \sim L(\nabla T \times M_z \hat{z})_x$, here $L$ being the channel length for each pair of electrodes pair that are separated along the $x$-direction, and $M_z$ the out-of-plane component of the induced magnetization in ML-WTe$_2$. The induced $M_z$ should follow that of the CGT surface layer, the source of the induced ferromagnetism. The slanted $V_{ANE}$ loops with nearly vanishing remanence at $H_z = 0$ are results of multidomain formation which produces stochastic responses in each single field sweep. The domain formation was previously studied in the anomalous Hall and magnetic force microscopy work of CGT thin flakes[31]. Linear heating power dependence of $V_{ANE}$ magnitude is consistent with the expected linear $\nabla T$-dependence (see Supplementary Sections 3 and 4).

To further investigate the correlation between the magnetic order in CGT and the induced ferromagnetism in ML-WTe$_2$, we carry out temperature dependence measurements of $V_{ANE}$. Figure 1e (and more data in other channels in Supplementary Section 5) shows the loops of normalized ANE voltage by power $P$, $V_{ANE}/P$, at selected temperatures. Because of the slanted loop shape, we take the saturation values of $V_{ANE}/P$ on both positive and negative field and plot the half difference between them in Fig. 1f for the three pairs. They all decrease as the device warms up before disappearing as the sample temperature $T_s$ approaches 61 K, the $T_c$ of CGT. $T_s$ is obtained by using the calibrated WTe$_2$ sample resistance as a sensitive thermometer. It is clear that the ANE signal is closely related to the ferromagnetic order parameter of CGT.

After experimentally establishing the induced ferromagnetism in ML-WTe$_2$, we turn to magneto-transport properties of the magnetized ML-WTe$_2$. By passing an AC current in ML-WTe$_2$ with the frequency of $f$ (13 Hz) and the root-mean-square (rms) amplitude of $I_{rms}$ (3 μA), we simultaneously measure the first- and second-harmonic longitudinal voltage responses, i.e., $1f$ and $2f$ voltages (measurement geometry sketched in Fig. 2a). Figure 2b plots the $H_z$-dependence of the $1f$ and $2f$ voltages from the 4–7 channel measured at 4 K. The raw $1f$ signal (top panel in Fig. 2b) clearly contains both $H_z$-symmetric ($V_{4-7}^{1f-S}$) and $H_z$-antisymmetric ($V_{4-7}^{1f-AS}$) components with comparable magnitude. The large $H_z$-antisymmetric signal mixed in the longitudinal channel is caused by the irregular WTe$_2$ shape, which is not etched into a regular Hall bar to avoid possible damages. After symmetrization and antisymmetrization of the longitudinal voltage, we obtain $V_{4-7}^{1f-S}$ and $V_{4-7}^{1f-AS}$ hysteresis loops that are characteristic of the AMR and AHE responses of ferromagnetic conductors, respectively (as shown in middle panel of Fig. 2b). As discussed in Supplementary Section 6, we exclude the spin Hall magnetoresistance mechanism that could arise from strong SOC in ML-WTe$_2$. Therefore, the magnetoresistance loops are possible only when the ML-WTe$_2$ layer is proximity-magnetized. Additionally, the $1f$ $H_z$-antisymmetric or the AHE signal in the longitudinal resistance channel provides another proof of the induced ferromagnetism in ML-WTe$_2$. In the Hall channel (between electrodes 7 and 9), we observe similar AHE signal mixed with a $H_z$-symmetric AMR signal but no observable quantized Hall signal is present. Similar proximity-induced AHE has also been observed in other quantum materials including graphene[23] and topological insulators[25,32].

Besides these $1f$ hysteresis loops, there is also a large $H_z$-antisymmetric hysteresis loop in the $2f$ response (Fig. 2b-bottom), which

has the same but inverted shape as the $1f$ AHE loop. In YIG/Pt heterostructures, passing a large AC current in Pt can indeed produce both $1f$ and $2f$ responses, but normally the former dominates;[33] and the $2f$ signal is attributed to SSE due to the Joule heating generated by the AC current in Pt. Similar to the nonlocal heating case, here the SSE mechanism by the sample self-heating induced an $2f$ out-of-plane $\nabla T$ component can also be excluded with the same reasons (see discussion in Supplementary Section 2). However, the sample self-heating generates an AC temperature rise in the ML-WTe$_2$ flake with the frequency of $2f$. The net outward heat dissipation into the surrounding medium results in a $2f$ in-plane $\nabla T$ component around the edge of the flake (see our COMSOL simulation results in Supplementary Section 4). This lateral $\nabla T$ component at the lower boundary is orthogonal to the longitudinal voltage channel as shown in Fig. 2a, and thus produces an ANE-like $2f$ response that is sensitive to $M_z$. In fact, the $2f$ signal has the same polarity as that of the ANE signal produced by the nonlocal heater in Fig. 1d with the same in-plane $\nabla T$ direction, which is consistent with the sample self-heating scenario. Hence, we attribute the $2f$ hysteresis to the ANE response to the sample Joule heating. Because of the hysteretic characteristic of the AHE responses, the $2f$ voltage is clearly unidirectional or nonreciprocal, depending on the direction of the magnetization. The nonreciprocity in $2f$ voltage may share the same origin with the nonreciprocal transport phenomena reported in other systems [27,34–36].

As both $1f$ and $2f$ signals are all consistent with the induced ferromagnetism in ML-WTe$_2$, they should vanish when CGT turns paramagnetic. Figure 2d–f show the respective $H_z$-dependence of AMR, AHE and ANE signals at selected temperatures. Indeed, they all disappear at and above the $T_c$ of CGT (as summarized in Fig. 2c).

Even though the lower electrodes in device D1 are placed close to the sample edge (as shown in Figs. 1c and 2a), they also contact the sample interior due to the transfer resolution limit; therefore, the measured signals contain contributions from both edge and bulk components. The pristine ML-WTe$_2$ is known to be a QSHI with gapless edge states; we also expect the magnetized ML-WTe$_2$ to contain both bulk and edge states. To understand their respective contributions, we fabricate another type of devices represented by D7 as illustrated in Fig. 3a, in which one set of multiple electrodes (from #3 to #6 in Fig. 3b) make electrical contact only with the bulk, called the Bulk-only electrodes, but the other set (from #11 to #14 in Fig. 3b) with both the bulk and edge, called the Bulk + Edge electrodes. The Bulk-only electrodes are electrically insulated from the sample edge using an additional thin layer of BN (~35 nm thick) to cover the pre-patterned Pt electrodes except for their far ends before transfer of ML-WTe$_2$ so that the uncovered ends of the Pt electrodes can only probe the bulk channel of ML-WTe$_2$. With these two sets of electrodes, we first measure the two-terminal resistance as a function of temperature for the Bulk-only and the Bulk + Edge electrodes, i.e., $R_{5-6}$ and $R_{13-14}$, respectively. Figure 3c reveals a striking contrast between these two sets. First, $R_{5-6}$ (Bulk-only) is always greater than $R_{13-14}$ (Bulk + Edge) over the entire temperature range, indicating that the edge is more conductive than bulk. Second, while the difference between them remains relatively small at high temperatures, the Bulk-only resistance $R_{5-6}$ rises steeply below 10 K while the Bulk + Edge resistance $R_{13-14}$ only shows a moderate increase. This is a sign of bulk carrier freezing indicative of a bulk gap. From the temperature dependence, we determine activation energy of 3.16 meV (Supplementary Section 7). This value is comparable to that found in CrI$_3$/WTe$_2$ (2.5 meV)[27] but much smaller than the QSHI bulk gap in ML-WTe$_2$ grown on a bilayer graphene substrate (~45 meV)[17]. Compared to pristine QSHI, magnetized QSHI is expected to exhibit a smaller gap due to spin splitting. For example, the spin splitting in conduction bands is about 30 meV for CGT/WTe$_2$ according to our density functional theory calculations (see Supplementary Section 8). We also note that a wide discrepancy in the band gap exists in the literature among various theoretical and experimental

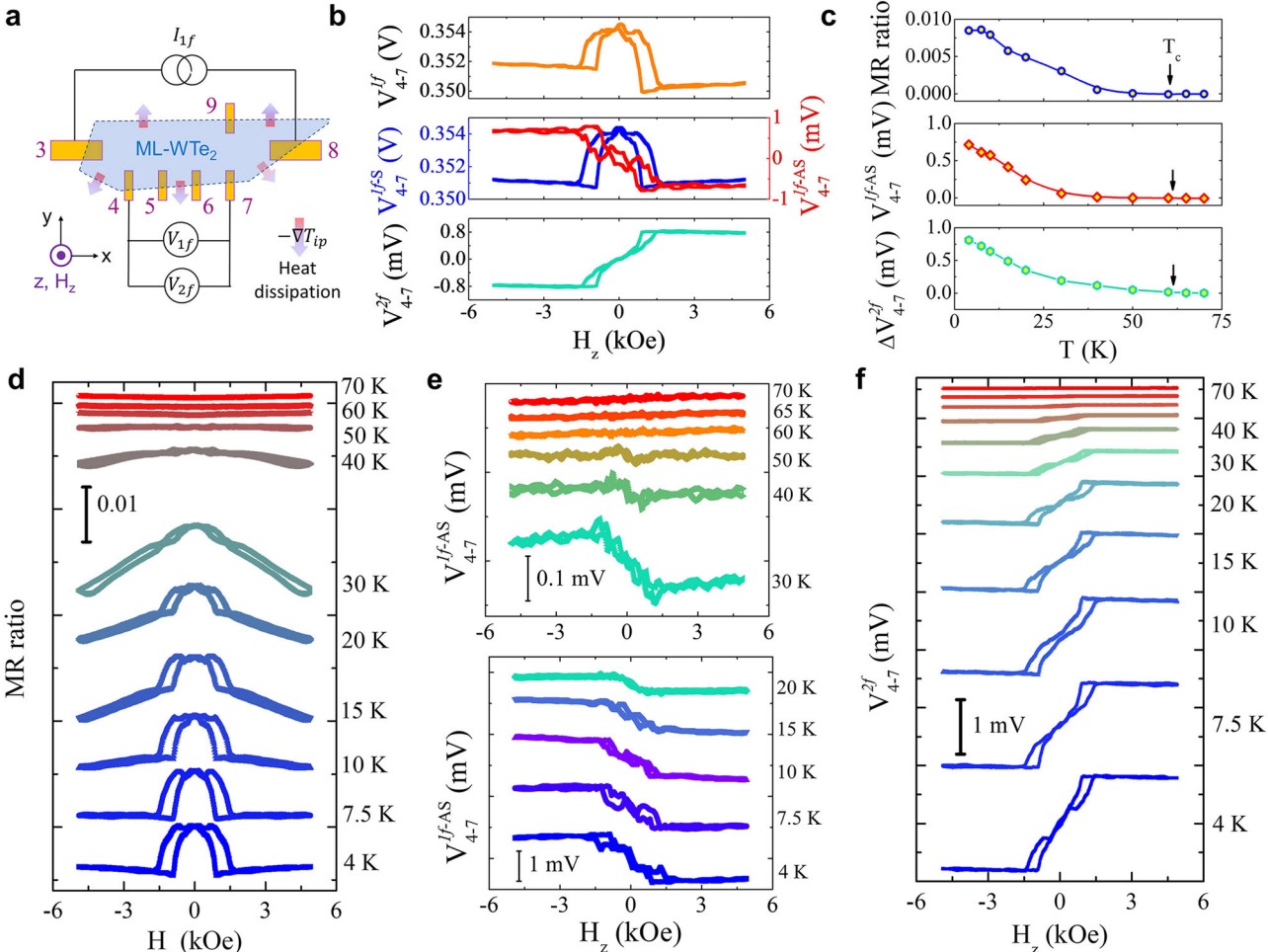

**Fig. 2 | Temperature dependence of linear and non-linear responses in ML-WTe₂/Cr₂Ge₂Te₆ heterostructure. a** Schematic illustration of measurement geometry. An AC current of 3 $\mu$A flows in the ML-WTe₂ flake through channel 3–8 (x-direction), and the $1f$ and $2f$ voltage responses are measured from channel 4-7 in sweeping $H_z$ with $f = 13$ Hz here. The thick arrows with color gradient across the device boundary indicate the direction of heat dissipation, i.e., $-\nabla T_{ip}$, where $\nabla T_{ip}$ is the in-plane component of the temperature gradient. **b** Magnetic field dependence of the raw $1f$, $V_{4-7}^{1f}$ (top), $1f$ $H_z$-symmetric, $V_{4-7}^{1f-S}$ and $H_z$-antisymmetric, $V_{4-7}^{1f-AS}$

(middle), and $2f$ $V_{4-7}^{2f}$ voltage (bottom) at 4 K. **c** Temperature dependence of the magnetoresistance (MR) ratio (top), AHE voltage (middle) and $2f$ voltage (bottom) taken from the data in **d**–**f** below. MR ratio magnitude is defined as [$V_{4-7}^{1f}$(5 kOe)-$V_{4-7}^{1f}$(0 Oe)]/$V_{4-7}^{1f}$(0 Oe), and $\triangle V_{4-7}^{2f} = V_{4-7}^{2f}(H_s) - V_{4-7}^{2f}(0)$. $H_s$ is the saturation magnetic field of $V_{4-7}^{2f}$. **d**–**f** Magnetic field dependence of MR ratio (**d**), $V_{4-7}^{1f-AS}$ (**e**) and $V_{4-7}^{2f}$ (**f**) voltage at various temperatures. The temperatures denoted here are read from the thermometer of the measurement system and the actual sample temperature should be higher due to local heating.

groups[15–17,27]. On the other hand, $R_{13\text{-}14}$ is essentially dominated by edge conduction below 10 K where the bulk carriers freeze out. In the meantime, $R_{13\text{-}14}$ approaches 303.7 k$\Omega$ at 2 K, which is over two order smaller than $R_{5\text{-}6}$ but at least a factor of 10 larger than $h/e^2$ (=25.8 k$\Omega$), the quantized resistance for a single 1D ballistic edge channel. This large edge resistance, which corresponds to an order of magnitude smaller than the quantized edge conductance, was also previously reported in similar heterostructures[27] or pristine ML-WTe₂[21] and explained by an edge state gap or backscattering in the edge channels. The large resistance also excludes the possibility of any charge transfer-induced conduction in CGT, which would decrease the resistance instead. We also perform gate voltage $V_g$ dependence measurements of the Bulk + Edge resistance. It generally shows a broad and shallow maximum centered around $V_g = 0$ at 4 K (Supplementary Section 9), which is likely due to the overlap between the small edge gap and bulk valence band.

Because of the small bulk gap, we need to stay below 10 K in temperature to access the QSHI edge transport by the Bulk+ Edge electrodes. To avoid sample heating and thus achieve lower sample temperature $T_s$ than in Figs. 1 and 2, we use a much smaller AC current ($I_{rms} = 0.5$ mA) in the nonlocal heater and simultaneously record ANE

voltages from the Bulk +Edge[13,14] and Bulk-only[5,6] electrodes. Figures 3d, e show the $T_s$-dependent ANE voltages from the Bulk + Edge and the Bulk-only electrodes scaled by the heating power $P$ of the nonlocal heater. The following surprising contrast stands out from the side-by-side comparison. First, the edge ANE signal does not vanish at low temperatures as one would expect for ideal 1D edge channels, but rather increases in its magnitude. Second, the edge ANE changes the sign as $T_s$ decreases, whereas the bulk ANE remains the same sign, highlighting distinctly different characteristics between the edge and bulk channels. To compare with the nonlocal heating case, we also deliberately inject smaller AC currents in ML-WTe₂ to reduce the self-heating power in order to achieve lower sample temperatures. Interestingly, the opposite signs between 4 K (Fig. 3f, left) and 14 K (Fig. 3g, left) of the $2f$ signals in the Bulk+ Edge electrodes also indicate a sign change between these two temperatures. In contrast, the $2f$ signals in the Bulk-only electrodes remain the same sign (Fig. 3f, g, right). The correspondence between the nonlocal- and self-heating cases further supports that the nonreciprocal $2f$ responses originate from ANE.

With the resistance and ANE data from the two sets of electrodes, we can readily separate the bulk and edge contributions using a simple two-component circuit model (as sketched in Fig. 4a, see details in

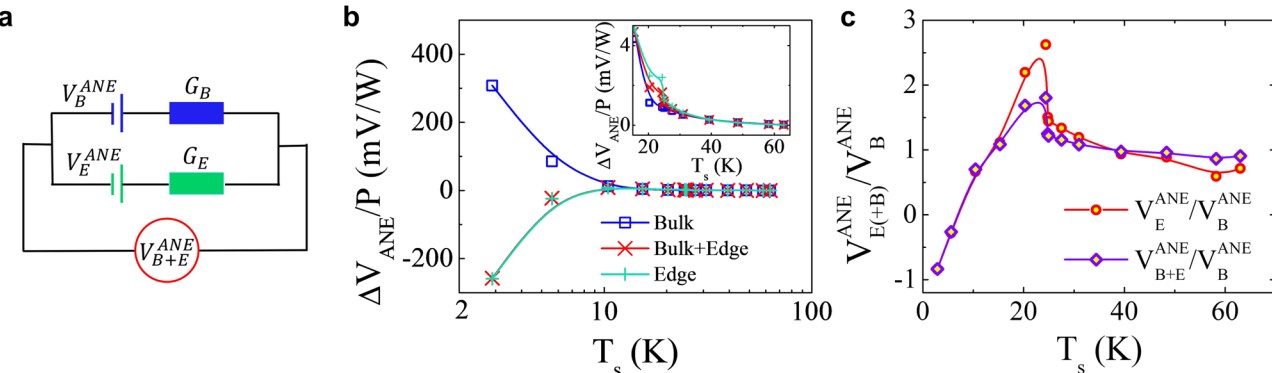

**Fig. 3 | Anomalous Nernst effect and anomalous Hall effect from edge and bulk channels of ML-WTe₂/Cr₂Ge₂Te₆. a** Schematic diagram of device structure. Electrodes on the left side probe combined edge and bulk signal of ML-WTe₂, and those on the right side are partly covered with BN to prevent edge contact with ML-WTe₂, thus only detect the bulk ANE. **b** Optical image of device D7 prior to transfer of ML-WTe₂/Cr₂Ge₂Te₆ composite layer. The ML-WTe₂ flake is indicated by the purple dashed polygon. Electrodes from 11 to14 probe the ANE signal from both edge and bulk, and electrodes from 3 to 6 probe the bulk-only ANE signal. The scale bar is 10 μm. **c** Temperature dependence of

resistance from 5–6 (Bulk) and 13-14 (Bulk + Edge). **d, e** ANE signals from 13–14 (**d**) and 5–6 (**e**) at selected temperatures. For $T_s < 20$ K, 40 loops are used for averaging; for $T_s > 20$ K, 20 loops are used for averaging. **f, g** 2$f$ current response $f$rom Bulk + Edge (11–14 and 11–12) and Bulk-only (5–6 and 5-4) channels to AC voltage applied between the two outermost horizontal electrodes on ML-WTe₂ vs. $H_z$ at $T_s = 4$ K (**f**) and 14 K (**g**). 60 loops are used for averaging in **f**, and 10 loops for other curves than the 11–14 electrodes (3 loops) in **g**, respectively. The rms magnitudes of the AC voltage for 4 K and 14 K measurements are 20 and 200 mV, respectively.

**Fig. 4 | Two-component transport from edge and bulk channels of ML-WTe₂.** **a** Illustration of the "parallel battery-resistor" model with ANE voltages from Edge ($V_E^{ANE}$) and Bulk ($V_B^{ANE}$) channels. $G_E(G_B)$ is the conductance of the Edge (Bulk) channel. $V_{B+E}^{ANE}$ is the total ANE signal from both edge and bulk. **b** Temperature

dependence of the ANE signal from Bulk, Bulk + Edge, and Edge channels. The Edge ANE is calculated using the "parallel battery-resistor" model. Inset shows a zoom-in plot of the high-temperature data. **c** Ratio of ANE from Edge (Bulk + Edge) channel to that from Bulk-only channel as a function of $T_s$.

Supplementary Section 10). Figure 4b plots the temperature dependence of ANE signals from bulk and edge channels, the latter of which is calculated using the circuit model. Clearly, both edge and bulk ANE signals increase in magnitude as $T_s$ approaches zero. This apparent low-temperature magnitude increase in both channels is possibly caused by the rapidly decreasing thermal conductivity due to phonon freezing, which greatly enlarges the actual $\nabla T$ (Supplementary Section 11). Since both channels are equally affected by the same thermal conductivity, we plot the ratio of the edge to bulk ANE voltages in Fig. 4c. Interestingly, aside from the sign change, the ratio shows a quick dive at low-$T_s$, which may be caused by the faster decreasing bulk ANE due to carrier freezing.

After successfully disentangling the two-component transport behavior, now we discuss implications of these observations. It was previously known that in channels longer than 100 nm[21,22], the smaller than quantized 1D conductance indicates that the QSHI edge states suffer from backscattering possibly due to inhomogeneous bulk states[37,38] and thus the 1D Dirac edge transport is diffusive. Here we propose an edge transport picture (see Fig. S9 in Supplementary Note 8) to explain the observed edge AHE signal (this mechanism can be modified to explain the edge ANE signal driven by a temperature gradient). Under broken time-reversal symmetry, the two counter-propagating helical Dirac edge states in magnetized QSHI do not have the same conductivity. In fact, the unquantized AHE must arise from a net charge current composed of two unequal counter-propagating flows of charge carriers, or a spin-polarized current. Under a voltage bias, the two longitudinal edges do not conduct the same amount of current, which leads to a net transverse or Hall voltage. The same inequivalent longitudinal edges, if they acquire energy-dependence in the conductivity due to the hybridization of the Dirac edge states with the bulk and/or ordinary edge states, also produce a net transverse voltage under a temperature gradient, which is the Nernst voltage. Such hybridization leads to quasi-1D edge states. The polarity of the voltages depends on the direction of the magnetization, which explains the observed $2f$ hysteresis. If the temperature varies, the chemical potential can intercept both the Dirac and ordinary edge states. Because the asymmetry between the both edge types can be different, it can result in different voltage responses. Although the actual outcome depends on the band structure and scattering details, the competition between the two types of edge states can in principle lead to a sign change in the transverse voltages. The low-temperature $2f$ sign change observed in our experiment may be a result of this competition.

In summary, we have demonstrated proximity-induced ferromagnetism in ML-WTe$_2$ using the vdW heterostructure approach. In this magnetized QSHI, we have unequivocally disentangled the edge and bulk transport from local transport probes. The nonzero edge AHE and ANE responses indicate that the edge states of the magnetized QSHI are partially spin-polarized, qualitatively different from the 1D ballistic chiral edges in QAHIs or helical edges in QSHIs.

## Methods

### Device fabrication

We fabricate our devices using the following processes (from bottom to top)[1]. The few-layer graphene (FLG) gate is transferred (with the BN using Polycarbonates pickup procedure) on a SiO$_2$/Si substrate with prepatterned Cr(5 nm)/Au(30 nm) heater using electron beam lithography (EBL)[2]. A thin hexagonal boron nitride (BN) layer (~35 nm) is transferred to electrically isolate the FLG gate and the pre-patterned heater from the monolayer (ML) 1 T' WTe$_2$ to be transferred on top[3]. Pt-electrodes are patterned on the top of the bottom BN using EBL and lift-off[4]. For devices like D1, a suitable ML-WTe$_2$ flake is identified under optical microscope (as shown in Supplementary Fig. S1a, b) and picked up using a thin flake of CGT, then the ML-WTe$_2$/CGT heterostructure as a composite layer is transferred onto the pre-patterned Pt-electrodes (as shown in Supplementary Fig. S1d). The WTe$_2$ bulk crystals in our experiments are purchased from 2D Semiconductor and the CGT crystals are provided by Prof. S. Jia's group at Peking University[5]. For device D7 with the edge insulated from the ML-WTe$_2$ interior (bulk) channel, a thin BN layer is transferred to cover the Pt electrodes except the far ends (Supplementary Fig. S1e) prior to the transfer of ML-WTe$_2$/CGT composite layer on to the pre-patterned Pt electrodes[6]. The ML-WTe$_2$/CGT heterostructure device including Pt electrodes is encapsulated by a top BN layer to prevent degradation and maintain device stability. The above exfoliation and transfer processes are carried out in a glove box in the atmosphere of argon to avoid device degradation. We have fabricated and studied seven devices in this work and they all show the same qualitative transport behaviors.

### Anomalous Nernst effect (ANE) measurements using nonlocal heater

An AC current with the frequency of 13 Hz is fed to the nonlocal heater using Keithley 6221 current source, and the second harmonic ($2f$) ANE voltage from different channels of ML-WTe$_2$ is detected using the standard lock-in technique. A magnetic field is swept perpendicular to the layers of the devices. All measurements for devices are performed using either a homemade closed-cycle system or Quantum Design's DynaCool Physical Property Measurement System (PPMS).

### Measurements of $1f$ and $2f$ responses to AC current in ML-WTe$_2$

We perform the linear ($1f$) and non-linear ($2f$) voltage measurements by feeding an AC current (voltage) into ML-WTe$_2$ of device D1 (D7) using Keithley 6221 current source (lock-in amplifier SR830), and $1f$ and $2f$ voltage (current) signals are probed using the standard lock-in method with the frequency of 13 Hz or 11 Hz. All measurements for devices are performed using either a homemade closed-cycle system or Quantum Design's DynaCool PPMS.

### Reporting summary

Further information on research design is available in the Nature Research Reporting Summary linked to this article.

## Data availability

The datasets generated during and/or analysed during the current study are available from the corresponding author on reasonable request.

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

## Acknowledgements

We thank Tang Su for his technical assistance at the beginning of the project, and Ady Stern, Roger Lake and Richard Wilson for helpful discussions. J.X.L. M.L. and J.S. acknowledge the support by DOE award #DE-FG02-07ER46351 and NSF-ECCS-2051450. M.R. and Y.-T.C. acknowledge the support from the NSF award DMR-2004701. B.Y. Acknowledges the financial support by the European Research Council (ERC Consolidator Grant "NonlinearTopo", No. 815869). K.W. and T.T. acknowledge support from the Elemental Strategy Initiative conducted by the MEXT, Japan (Grant Number JPMXP0112101001) and JSPS KAKENHI (Grant Numbers 19H05790, 20H00354 and 21H05233).

## Author contributions

J.S. conceived the idea, designed the experiments and wrote the paper. J.X.L. and M.R. carried out the main experimental work supervised by J.S. and Y.T.C., respectively. M.L. participated in the early experimental work. J.Y.K. performed band structure calculations supervised by B.H.Y. Y.M.X. performed COMSOL calculations supervised by X.C. X.Z. grew the CGT crystals supervised by S.J. K.W. and T.T. provided BN crystals. J.X.L., B.H.Y. and Y.T.C. contributed to the data analysis and interpretation and manuscript preparation.

## Competing interests

The authors declare no competing interests.
