## [Peer Review File · Nature Communications]

Reviewers' Comments:

Reviewer #1:

Remarks to the Author:

Li et al reported anomalous Nernst effect (ANE) and non-reciprocal anomalous Hall effect (AHE) in monolayer 1T' WTe₂/Cr₂Ge₂Te₆ heterostructure. Their major conclusion is that proximity to Cr₂Ge₂Te₆ induced a ferromagnetic quantum spin hall insulator state in monolayer 1T' WTe₂, which the authors seem to suggest is different from a quantum spin hall insulator or a quantum anomalous hall insulator.

Although I'm not sure what role the AHE is playing in this manuscript, I think ANE in such a van der Waals heterostructure could be an interesting result. The problem is the conclusion the authors draw from their observation. I'm most confused by the notion of a ferromagnetic quantum spin hall insulator, a name which the authors create but do not define in the manuscript. I also could not find any clear and convincing evidence for their claim that this is different from a quantum spin hall insulator or a quantum anomalous hall insulator. How does the ANE data support this claim? Nor could I find any concrete picture of the physics involved. It looks to me that the authors just put together some experimental results they could not quite explain and gave a name to it. For these reasons, I do not recommend publication of this manuscript.

The following are some more technical comments.

1. Although Cr₂Ge₂Te₂ is a ferromagnetic insulator which does not have conducting electrons, its magnetic moment may still produce a stray magnetic field to affect WTe₂'s transport signal. We noticed that previous works (Reference 26 and 27) have investigated proximity effect between monolayer WTe₂ and layered antiferromagnetic CrI₃. In those works, the antiferromagnetic state of CrI₃ excluded the effect from stray magnetic field and conclusively showed exchange interaction. But Cr₂Ge₂Te₂ does not have an antiferromagnetic state to make such a discrimination.
2. The ferromagnetic ordering in WTe₂ was not established. Just because the ANE of the heterostructure showed magnetic field dependence does not mean there is ferromagnetic ordering in WTe₂. The field dependence could come from Cr₂Ge₂Te₂.
3. Related to the above point, the critical temperature of this supposedly induced ferromagnetic state in WTe₂ is the same as the Curie temperature of Cr₂Ge₂Te₂. This is highly unlikely. Proximity-induced ordering usually has a much lower critical temperature than the host material.
4. There is a mistake on Line 136 and 137 in the supplementary about the Bulk+Edge channel (which should be 13-14) and the Bulk-only channel (which should be 5-6).

Reviewer #2:

Remarks to the Author:

This paper reports the transport in WTe₂/CGT heterostructure, where CGT is a FM semiconductor and monolayer WTe₂ is presumably a quantum spin Hall insulator. The magnetic proximity effect from CGT was proposed to break the TRS and induce ferromagnetism in WTe₂. ANE and AHE was provided as evidence for the proximity effect. Edge transport and bulk transport was separated using local electrodes. This work follows the groups' other works on transport studies of proximity effects in different systems. I find the separation of edge and bulk transport, which shows clearly contrasting behavior very interesting. However, I am not ready to be convinced that the edge transport is of quantum nature. There are also other inconsistencies. I suggest a major revision of the present manuscript.

Please see my specific comments below:

1. It is clear that the transport dominated by the edge and bulk are different, with edge being more conducting. However, this is the only conclusion that can be drawn. The 1D edge channel maybe involved. However, it is mixed with bulk transport, and there is no evidence of the quantum nature.

2. Moreover, could this difference come from the inhomogeneity in different parts of the sample, making edge more conducting? The authors did not provide structural characterizations of monolayer WTe₂. Also, how many samples have they measured? Do they all show the same qualitative behavior?
3. ANE and AHE show slanted loop with 0 remanance, which was attributed to multi-domain. However, thin layers with perpendicular anisotropy are expected to show high remanance, e.g. in case of CrI₃ or even MBE grown CGT (APL Mater 6, 091104). It is not clear in few layer CGT, whether the easy axis is even out of the plane, or closer to Heisenberg-type. The authors themselves seem to make contradictory arguments, saying lack of in-plane magnetization (line 96) and developing in-plane components (line 138). However, from the hysteresis it is clear in-plane magnetization is present, invalidating partly the claim of lacking of ISSE or ISHE, especially if WTe₂ is not an ideal 2D system.
4. The way the authors define edge and bulk is vague. Given that the device is made at the micrometer scale, it is not certain whether bulk and edge can be differentiated.
5. One of the key piece of data, Fig. 4b and text L.227, showed an increasing thermoelectric signal when T approaches zero. This is non-physical! All thermoelectric response goes to zero when T approaches zero by the 3rd law of thermodynamics. This raises severe concern about their measurements (see also points below for a related tech concern).
6. When excluding SSE as the source of the observed thermoelectric response, the author did not discuss the possible out-of-plane thermal gradient; also, the use of Pt as electrodes further complicate the experiment, as the ISHE in Pt is known to be strong and could potentially affect the measurement.
7. The Pt or Cr/Au electrodes which also serve as thermometry in the experiments typically do not function when T < 20 K, a temperature below which the resistivity of these electrodes saturates. I do not understand how they manage to overcome this problem.
8. What additional insights does ANE provide other than complementing the AHE data?

Reviewer #3:

Remarks to the Author:

The authors manuscript focuses on nonreciprocal responses (with respect to the magnetization orientation) in Van der Waals heterostructures. Magnetization is induced in the sample by bringing a few-layer insulating ferromagnet (Cr₂Ge₂Te₆) close to the heterostructure (WTe₂). As such, the resulting system may host anomalous Nernst effect (ANE), anomalous Hall effect (AHE), and anisotropic magnetoresistance effect (AMRE), all of which were measured and reported by the authors. The authors then go on to analyze their results distinguishing between bulk and edge responses of anomalous Nernst effect and AHE.

While I find the manuscript interesting, there are significant deficiencies which give me pause in recommending it (see major comments below: e.g., authors claim an AHE signal in Figure 2, but it is unclear from their schematic in Fig. 2a that it is indeed of a "transverse" Hall nature. They seem to drive current between contacts 3-8, and then measure 4-7; while antisymmetrization may pick out signals that are antisymmetric, how is this transverse, how does it show AHE?).

Another major place for pause is that I do not sufficiently understand what conclusion is new here in this manuscript as compared with what has been done previously, e.g., Ref. 27. In Ref. 27, a magnetic proximity effect was seen in WTe₂ (albeit using CrI₃). What does the present manuscript uncover about physics that is different (it seems that Ref. 27 already had nonreciprocal magnetoresistance). Is it Nernst effect and Hall signals (if so, what new physics does this reveal)? Is it the new magnetic substrate of CrGeTe (if so what new physics does this tell us about, how is this different from CrI₃)? Is it the difference between bulk and edge contributions (if so, what new physics does this teach us?). It would be helpful for the reader to know what is the value proposition of this new paper.

As a result, I do not recommend this paper for publication in its current form. Additionally, while the data is certainly interesting, unfortunately, I found the manuscript a little haphazardly written.

Major comments:

1. In the paper, much attention focussed on AMRE in the heterostructure. However, similar results are obtained and reported in Ref. 27. What are the main differences between the Author's results and Ref. 27, apart from using CrGeTe in the heterostructure (as opposed to CrI₃)? Additionally, I note that Ref. 27 also discussed a possibility of ANE as a mechanism for their non-reciprocity (see e.g., second last paragraph of Ref. 27 main text).

2. How do I interpret (any of) the results of Fig. 2 as an AHE? it seems that current was driven between 3-8, and then voltage was then measured between 4-7. Aren't all these contacts longitudinal? What is "Hall" about them? It is not clear that antisymmetrization necessarily captures "Hall" (see top of page 5 of text); if it does, the authors should thoroughly justify.

3. It would seem that a prominent messages of the paper is the claim regarding edge states, last line of main text: "The nonzero edge ANE responses indicate that the ferromagnetic QSHI edge states are qualitatively different from the 1D ballistic chiral or helical edge states." The result seems to be too rapidly generalized and too vaguely discussed. For instance, if one has a quantum Hall system with chiral edge states, wouldn't one expect a nernst effect? Similarly, wouldn't a QAHE system also display nernst signals as well? Why is the comparison with 1D even useful?

Minor comments:

1. The paper has a strange organization. The analysis starts from the discussion of ANE, then shifts to the debate on AHE and AMRE and then returns to the discussion of ANE. This format of narration was harder to follow, including the fact that the results and discussion of FIG. 2 seem to repeat the message of Ref. 27, I would recommend modestly restructuring the paper to make clear what the main point of the paper is.

2. The blue curve plot on FIG. 1f has a strange non-monotonicity at low temperatures. However, this was not commented on in the paper. Clarification of what can be the reason for such behaviour would be beneficial.

3. FIG 4b is missing a curve that corresponds to the "Bulk+Edge contribution" in the main panel. Is it only in the inset? If it is only in inset, why?

4. On page 8: authors write "However, a strictly 1D system should not produce any diffusive transverse transport responses such as AHE or ANE signals" What do they mean? It is clear that the system they are studying is not 1D since it is 2D WTe₂ on CrGeTe, so what is the purpose of this statement.

REPLY TO REVIEWER COMMENTS

We thank the reviewers for the constructive comments (in blue) and will address them point-by-point below (in black).

Comments:

Reviewer #1 (Remarks to the Author):

Li et al reported anomalous Nernst effect (ANE) and non-reciprocal anomalous Hall effect (AHE) in monolayer 1T' WTe₂/Cr₂Ge₂Te₆ heterostructure. Their major conclusion is that proximity to Cr₂Ge₂Te₆ induced a ferromagnetic quantum spin hall insulator state in monolayer 1T' WTe₂, which the authors seem to suggest is different from a quantum spin hall insulator or a quantum anomalous hall insulator.

Although I'm not sure what role the AHE is playing in this manuscript, I think ANE in such a van der Waals heterostructure could be an interesting result. The problem is the conclusion the authors draw from their observation. I'm most confused by the notion of a ferromagnetic quantum spin hall insulator, a name which the authors create but do not define in the manuscript. I also could not find any clear and convincing evidence for their claim that this is different from a quantum spin hall insulator or a quantum anomalous hall insulator. How does the ANE data support this claim? Nor could I find any concrete picture of the physics involved. It looks to me that the authors just put together some experimental results they could not quite explain and gave a name to it. For these reasons, I do not recommend publication of this manuscript.

Reply: These are valid criticisms and we thank the reviewer for pointing them out.

First of all, we agree that we need to streamline the presentation to make the points more easily crossed to readers in a broader community.

"Ferromagnetic quantum spin Hall insulator" is just a label for the special insulator like WTe₂/CGT in order to draw the contrast with the quantum spin Hall insulator (QSHI) and the quantum anomalous Hall insulator (QAHI). It is simply a "magnetized quantum spin Hall insulator". To avoid any possible ambiguity, we now call it "magnetized quantum spin Hall insulator (MQSHI)" in the revised manuscript. In addition, we have changed the title to "Proximity-magnetized quantum spin Hall insulator: monolayer 1T' WTe₂/Cr₂Ge₂Te₆" to place an emphasis on the induced ferromagnetism in MQSHI.

Although QSHI, QAHI, and MQSHI all have insulating bulk and support metallic edge channels, there is an important difference: distinct spin polarizations in the charge carriers of the metallic edge. While the carrier spin polarization is zero in the QSHI and 100% in the QAHI, the MQSHI has spin polarization between zero and 100%, somewhat like conductive ferromagnets. However, our WTe₂/CGT is by no means an ordinary magnetic conductor or an ordinary magnetic insulator.

In short, a key difference among the QSHI, QAHI, and MQSHI lies in their edge states, i.e., whether the carriers in the metallic edges are spin polarized and how much if they are. We also

pointed out the spin-polarized characteristic in the abstract, introduction, and summary. We hope this clears the confusion to the reviewer and potentially to many readers.

Now about ANE. First of all, due to historical reasons, the word “anomalous” was given to the Nernst effect signal that shows an open hysteresis loop or saturation (e.g., may not show an open loop in the case of hard-axis responses) at strong magnetic fields, very much like the anomalous Hall effect. Both AHE and ANE contain a magnetization-dependent response. It is distinctly different from the ordinary Nernst effect which only depends on the applied magnetic field; thus, is not hysteretic. When ANE signals are observed, it generally entails that they are produced by a ferromagnetic source in the sample. The ANE can come from either a conductive ferromagnet or a non-magnetic conductor that is partly magnetized. It is true that the implication of ANE may not be appreciated by researchers outside the magnetism and spintronics communities. What is special here is that WTe₂/CGT has both edge and bulk channels. Then there is a natural question of which one is responsible for the ANE signals, which is what the second part of the manuscript aims to address. We disagree with the reviewer that our manuscript contains a random set of experimental data that can't be explained. As will be discussed more in our reply to the 3rd reviewer, these results are tied in together. However, we bear the responsibility of presenting them well, which is what we have striven to do in this revision. To make the presentation clear, now we moved some detailed discussions to Suppl. Info.

We gave the label “ferromagnetic QSHI” to WTe₂/CGT because of the following unique properties: (1) the only conductive component in the samples is the monolayer WTe₂ and the transport properties are directly from WTe₂; (2) when we observe ANE, AMR, and AHE, it is the WTe₂ that behaves as if it was a ferromagnet; and (3) edge channels are clearly present and the edge is partially spin-polarized. These characteristics are unique to WTe₂/CGT and we have tried to make these points clearer in the revised manuscript.

Comment: The following are some more technical comments.

1. Although Cr₂Ge₂Te₂ is a ferromagnetic insulator which does not have conducting electrons, its magnetic moment may still produce a stray magnetic field to affect WTe₂'s transport signal. We noticed that previous works (Reference 26 and 27) have investigated proximity effect between monolayer WTe₂ and layered antiferromagnetic CrI₃. In those works, the antiferromagnetic state of CrI₃ excluded the effect from stray magnetic field and conclusively showed exchange interaction. But Cr₂Ge₂Te₂ does not have an antiferromagnetic state to make such a discrimination.

Reply: As the reviewer correctly stated, Cr₂Ge₂Te₆ (CGT) is a ferromagnetic insulator and it does produce stray fields near the edge of the sample and at domain walls. The reviewer also pointed out the advantage of using CrI₃ to ascertain the proximity-induced effect by the surface spins. However, as it has been investigated quite extensively in magnetism, stray field alone in most ferromagnets does not produce large enough Hall resistance hysteresis. In general, the anomalous Nernst signal level at saturation is much greater (order(s) of magnitude) than the ordinary (i.e., the field-linear background signal) Nernst signal level at the saturation fields. It can also be seen in our samples in Figs. 1d and 1e and other figures. Similar arguments hold

for the anomalous Hall effect. For ferromagnets, the maximum stray magnetic field strength is $\sim 4\pi M_s$, which is only about 0.2 T for CGT. From Fig. 1d, the linear ordinary Nernst background over a ~ 2 T field range is at least one order smaller than the saturation signal; therefore, with a stray field jump of ~ 0.4 T, it can only produce an ordinary Nernst jump that is about 2 orders smaller than the saturation level. Clearly, the hysteresis of this magnitude cannot possibly be produced by the stray field hysteresis. In ferromagnets, the consensus is that the anomalous Hall effect is from the spin-orbit coupling and likewise, the large anomalous Nernst effect is from the same origin (both are fundamentally related). To better explain to Nat Comm readers in a broad community, we have added a brief explanation about the stray field effect in the first paragraph on page 4.

Comment: 2. The ferromagnetic ordering in WTe₂ was not established. Just because the ANE of the heterostructure showed magnetic field dependence does not mean there is ferromagnetic ordering in WTe₂. The field dependence could come from Cr₂Ge₂Te₂.

Reply: Based on our experimental observations (i.e., AHE, ANE, and AMR) and the explanation given above, we are confident that WTe₂ has induced ferromagnetism by CGT. These physical quantities depend on induced magnetization in WTe₂ by CGT. There are two main reasons for this. First, CGT resistance (> 10 G Ω) is more than 4 orders of magnitude larger than WTe₂ ($< M\Omega$) resistance below CGT's Curie temperature. So the current can only flow in WTe₂. Second, our exfoliation and transfer process is an inherently low-temperature one in which no diffusion or doping was found to cause CGT to turn to conducting in our previous study (M. Lohmann et al., Ref. 31). In fact, the >10 G Ω resistance in CGT flakes was quoted from Ref. 31. When we detect ferromagnetic-like responses, they must originate from the proximity-induced effects. The hysteresis in these transport quantities is expected to follow that of CGT. More will be discussed in the answer to the next question.

Comment: 3. Related to the above point, the critical temperature of this supposedly induced ferromagnetic state in WTe₂ is the same as the Curie temperature of Cr₂Ge₂Te₂. This is highly unlikely. Proximity-induced ordering usually has a much lower critical temperature than the host material.

Reply: This is a fair point. It is known that the applied magnetic field drives continuous ferromagnetic phase transition to 1st order. An established way of accurately determining the Curie temperature is by using the Arrott plot, which requires field dependence measurements over a range of fields and temperatures near the transition temperature. In our devices, we do not have sufficiently high signal-to-noise ratio in our ANE signal to allow us to extract the Curie temperature accurately using the Arrott plot as we did for FGT (M. Alghamdi et al., Nano Lett. **19**, 4400 (2019)). With the field-induced broadening of the critical transition in our experiments, we observed approximately the same critical temperature as that of CGT, which presents us from resolving a small difference between the two. It is possible that they only differ by several degrees.

With such a small difference, one may ask why the Curie temperature of the induced ferromagnetic phase is so close to that of CGT. While the exact reason is unclear, we can

imagine that the monolayer thickness of the WTe₂ flakes may play an important role. In vdW heterostructures, an advantage is to leverage the atomically flat layers to facilitate an intimate contact between different materials. Although the exchange coupling strength is short-ranged, if the interface is atomically flat so that one atomic layer is in the exchange range, the entire sample is magnetized by the exchange field. This is likely a common feature to vdW heterostructures involving a ferromagnet. In fact, we would like to point out that in the previous study of WTe₂/CrI₃ (Ref. 27), the observed transition temperature for the proximity effect is close to 45 K, which is the magnetic transition temperature of bilayer CrI₃.

Comment: 4. There is a mistake on Line 136 and 137 in the supplementary about the Bulk+Edge channel (which should be 13-14) and the Bulk-only channel (which should be 5-6).

Reply: We thank the reviewer for pointing this out. It is corrected in the revised manuscript.

Comment: Reviewer #2 (Remarks to the Author):

This paper reports the transport in WTe₂/CGT heterostructure, where CGT is a FM semiconductor and monolayer WTe₂ is presumably a quantum spin Hall insulator. The magnetic proximity effect from CGT was proposed to break the TRS and induce ferromagnetism in WTe₂. ANE and AHE was provided as evidence for the proximity effect. Edge transport and bulk transport was separated using local electrodes. This work follows the groups' other works on transport studies of proximity effects in different systems. I find the separation of edge and bulk transport, which shows clearly contrasting behavior very interesting. However, I am not ready to be convinced that the edge transport is of quantum nature. There are also other inconsistencies. I suggest a major revision of the present manuscript.

Reply: We thank the reviewer for reviewing our manuscript and providing the assessment.

Comment: Please see my specific comments below:

1. It is clear that the transport dominated by the edge and bulk are different, with edge being more conducting. However, this is the only conclusion that can be drawn. The 1D edge channel maybe involved. However, it is mixed with bulk transport, and there is no evidence of the quantum nature.

Reply: The reviewer is correct. There is no quantized conductance in WTe₂/CGT as one would expect from ideal 1D ballistic transport. It is even true for QSHIs including HgTe/CdTe and standalone WTe₂. This is a "drawback" of quantum spin Hall insulators (QSHIs) compared to quantum anomalous Hall insulators (QAHIs) in which only one chiral edge channel exists and backscattering is therefore eliminated. Nevertheless, the gapless edge channel transport in QSHIs has been clearly shown in refs. 21 and 22. Although the conductance in QSHIs does not take the quantized value, it remains finite as the Fermi level falls in the bulk band gap. The situation is similar here in magnetized QSHI. Despite that the conductance is not quantized, it is definitely not an ordinary insulator because the edge states conduct. We have found in this study that the edge channels in magnetized QSHI can have a finite spin polarization, which puts

itself in a different material category. The term "Ferromagnetic quantum spin Hall insulator" may be misleading, and we have changed it to "magnetized quantum spin Hall insulator", as addressed in our response to the 1st reviewer.

Comment: 2. Moreover, could this difference come from the inhomogeneity in different parts of the sample, making edge more conducting? The authors did not provide structural characterizations of monolayer WTe₂. Also, how many samples have they measured? Do they all show the same qualitative behavior?

Reply: If we understand correctly, the reviewer seems to suggest that the WTe₂ bulk may become insulating due to disorder in the middle of the sample. Although it is difficult to completely exclude this possibility, we did not target any specific part of the substrate during transfer. Therefore, we believe that it should not be the case. We fabricated seven devices as stated in the Methods part of the manuscript (6 D1-like devices as shown in Fig. 1c and 1 D7-like devices as shown in Fig. 3b). All devices indeed show the same qualitative bulk+ edge behaviors.

We have added our Raman spectroscopy data in Fig. S1 in SI to show the two characteristic modes for single atomic layer WTe₂. The measurements were taken with BN on top to protect WL WTe₂ from oxidation. We currently do not have the capability of doing AFM or STM imaging on pristine ML WTe₂ flakes inside the glovebox or ultrahigh vacuum.

Comment: 3. ANE and AHE show slanted loop with 0 remanance, which was attributed to multi-domain. However, thin layers with perpendicular anisotropy are expected to show high remanance, e.g. in case of CrI₃ or even MBE grown CGT (APL Mater 6, 091104). It is not clear in few layer CGT, whether the easy axis is even out of the plane, or closer to Heisenberg-type. The authors themselves seem to make contradictory arguments, saying lack of in-plane magnetization (line 96) and developing in-plane components (line 138). However, from the hysteresis it is clear in-plane magnetization is present, invalidating partly the claim of lacking of ISSE or ISHE, especially if WTe₂ is not an ideal 2D system.

Reply: We thank the reviewer for pointing out the 0 remanance observation. We could have explained better to avoid such a perception. We added sentences on page 4 to indicate that in Fig. 1, 20 loops were used to make the open loop collapsed. As the reviewer may have noticed, some hysteresis loops (e.g., in Fig. 1) have nearly zero remanance, while others (e.g., Fig. 3g) do show clear open hysteresis loops. So the loops do not always show zero remanance. In our data analysis, the zero remanance is actually caused by averaging multiple (10's) slanted loops in which the domain wall motion-induced changes are stochastic. When signal is relatively large as in Fig. 3g, taking a few field sweeps (as few as 3 in 11-14 electrode data) is enough. However, when signal is weak (nearly 100 times small) as in Fig. 3f, the loops are relatively noisy, and 60 loops are used to average in order to get decent ANE signals. Since individual loops can have large fluctuations, averaging of a large number of loops results in a collapse of open loops. This is also true in some cases (Fig. 1) where we intended to improve the data quality by performing 20-loop averaging. The domain wall nucleation and propagation was attributed to the slanted loop shape as reported previously by MFM and transport measurements in our previous work (Ref. 31). We also added a sentence near the end of the first paragraph on Fig. 4 to refer to the stochastic nature of the responses.

About the anisotropy, we know that CGT has much smaller anisotropy than FGT (10^5 erg/cc vs 10^7 erg/cc) (Ref. 28 on CGT and M. Alghamdi et. al.; Nano Lett. 19, 4400 (2019) on FGT). In both vdW materials, the anisotropy axis is perpendicular to the atomic planes. As we know in magnetism, the domain formation is largely driven by dipolar interaction that counteracts against anisotropy and exchange energies. As a result, in thicker layers the dipolar energy wins out and domains are favored. In relatively thick CGT, the loops are slanted and domains are observed (in Ref. 31). In FGT, however, due to its stronger anisotropy, even thick flakes (up to ~ 100 nm) show squared hysteresis loops. Therefore, the loop shape is an outcome of the competition of several energies which depends on the details of magnetic samples. In the figure caption, we have added the averaging and explained why some loops have zero remanence.

About the “contradictory arguments” pointed out by the reviewer, we believe that the misunderstanding arises from L96. We intended to say that the spin-charge conversion (spin Seebeck effect) by WTe₂ could only primarily reveal the M_x and M_y components – if the lateral temperature gradient dominates, but in the observed hysteresis loops under Hz fields, the signal resembles the M_z characteristics instead (both M_x and M_y would disappear at large Hz, but the observed signal increases and saturates at large Hz). Hence, this signal is not likely produced by the SSE effect (we concluded that it is from ANE under a lateral temperature gradient). This does not mean that M_x and M_y components do not exist when domains are present. On the contrary, M_x and M_y do exist but we do not observe their consequence if the SSE was the mechanism. We hope we explained clearly that the statement in L96 does not contradict with L138.

Comment: 4. The way the authors define edge and bulk is vague. Given that the device is made at the micrometer scale, it is not certain whether bulk and edge can be differentiated.

Reply: In principle, the edge current channel in WTe₂ is about ~ 100 nm wide (Ref. 18), which is below the resolution of the alignment process during transfer. To distinguish the bulk and edge contribution, we used a special device geometry in which an extra BN flake is inserted in between ML WTe₂ and Pt electrodes to prevent electrical contact to the WTe₂ edge. Fig. 3b caption and the Methods section briefly described the process. As shown in Fig. 3b, the upper boundary of the inserted BN flake well passes the WTe₂ lower boundary so that the edge channel (~ 100 nm) in WTe₂ is completely insulated from prefabricated Pt electrodes (from #3 to #6) by BN after the entire stack is transferred. These Pt electrodes (from #3 to #6) are only electrically connected to the bulk, but not at all to the edge. The “Bulk-only” data were taken with these electrodes. A similar method for probing bulk-only states was recently reported in Ref. 19. For the electrodes without the extra BN spacer (from #11 to #14), both edge and bulk are in electrical contact with the electrodes, which are used to obtain data for “Bulk+ Edge”. We believe that this geometry allows us to separate the edge from the bulk, as we see distinct behaviors, for example, changing signs at low temperature when the bulk becomes highly insulating.

Comment: 5. One of the key piece of data, Fig. 4b and text L.227, showed an increasing thermoelectric signal when T approaches zero. This is non-physical! All thermoelectric response goes to zero when T approaches zero by the 3rd law of thermodynamics. This raises severe concern about their measurements (see also points below for a related tech concern).

Reply: The reviewer is correct and we thank the reviewer for pointing this out. We explained this ostensible contradiction in Suppl. Info. Section 11. We hope the explanation we offered there can address reviewer's question about this low-temperature diverging trend.

Comment: 6. When excluding SSE as the source of the observed thermoelectric response, the author did not discuss the possible out-of-plane thermal gradient; also, the use of Pt as electrodes further complicate the experiment, as the ISHE in Pt is known to be strong and could potentially affect the measurement.

Reply: This is a good point. The out-of-plane component of the temperature gradient may exist or may even not be negligible. This would drive a spin current flow out of the plane. Similar to a previous discussion about the SSE contribution, the ISHE signal in SSE can't explain our data. The spin-charge conversion requires the "Hall" geometry, i.e., the spin polarization, spin current flow must not be collinear in order to yield finite ISHE signals. If gradT is out-of-plane, the ISHE signal would not show any Mz characteristics. On the contrary, the observed hysteresis loop is mainly characteristic of Mz and does not contain, at least within the experimental resolution, any Mx or My characteristic. Based on this analysis, we believe that even if there is an out-of-plane component in gradT, it still does not explain our field-dependent signals. The logical conclusion is that either the out-of-plane gradT is very small or the SSE is not operative here, as discussed in the text. We thank the reviewer for raising this possibility and modified a sentence in the last paragraph on page 3 by inserting "mainly" before "a lateral temperature gradient".

The reviewer is correct about the downside of using Pt as electrodes for possible complications. Ideally, Pt should be avoided. In this study, we wanted to prevent WTe₂ from being exposed to air, chemicals, and electron beam in the lithography process. Therefore, we adopted this special pickup/transfer process to place WTe₂/CGT/BN directly on to pre-patterned electrodes inside the glovebox. Pt is a suitable electrode materials because the surface does not easily get oxidized. Pt also makes good electrical contact to WTe₂, possibly due to its high work function, compared to other inert metals, such as Au. Nevertheless, since (1) Pt-WTe₂ contact area is relatively small compared to WTe₂-CGT contact, and (2) the ISHE signal from Pt in the small contact area, if any, would be very small across the narrow width (~ 1 μm) of the Pt electrodes; consequently, we do not expect a significant ISHE signal from Pt.

Comment: 7. The Pt or Cr/Au electrodes which also serve as thermometry in the experiments typically do not function when $T < 20$ K, a temperature below which the resistivity of these electrodes saturates. I do not understand how they manage to overcome this problem.
8. What additional insights does ANE provide other than complementing the AHE data?

Reply: The reviewer is correct about the shortcoming of using Pt or Cr/Au as a thermometer in general for low temperature measurements. We have made it clear in this revision that the sample temperature is calibrated using the sample resistance itself, i.e., WTe₂, not Pt or Cr/Au. At low temperatures, WTe₂ resistance is very sensitive to temperature so that the sample temperature can be accurately determined (as indicated in Fig. 3c). On the other hand, to measure the mean temperature difference between the heater (Cr/Au) and the Pt electrodes where the ANE signals are measured, we measured the resistance of the heater to calibrate the heater temperature. Before the heater is turned on, we take the system temperature as the heater temperature since the system is at thermal equilibrium. As the heater is turned on, the local temperature of the heater increases, the Cr/Au heater resistance is still sensitive to its

temperature. This does not affect the temperature reading for the heater site. In the self-heating measurements using small AC electrical currents in WTe₂, the sample temperature can reach as low as 4 K (Fig. 3f), but the WTe₂ thermometry can work perfectly. In this case, we do not know the actual temperature gradient. We appreciate the reviewer for pointing out the ambiguity about the resistance thermometry measurements. At the bottom of page 4, we clarified it by stating how T_s is measured.

About the additional insights provided by ANE, first of all, very much like AHE, the existence of ANE is a manifestation of ferromagnetism in a conductive sample or part of a conductor. In the WTe₂/CrI₃ (Ref. 27), the authors did not see any AHE mixed in the 1f (first-harmonic) longitudinal channel designed to detect edge currents. There were no Hall electrodes in those devices, so no AHE was reported. In our data, the 1f- channels (both R_{xx} and R_{xy}) contain a definitive antisymmetric, albeit small, component. This small signal alone would not be sufficient for us to confidently claim induced ferromagnetism. In contrast, the 2f-signals, measured with both the nonlocal heater and local current heating, unmistakably show large ANE hysteresis in the proper ANE geometry (i.e., voltages from electrodes perpendicular to gradT). In our study, ANE alone can allow us to ascertain induced ferromagnetism (after we argued against the SSE origin). If both AHE and ANE signals are large, ANE can provide some extra information about the band structure via the energy derivative of the anomalous Hall conductivity if it is due to the intrinsic effect or about the scattering if extrinsic. For some reason, the AHE signal is very small compared to ANE. We do not have enough experimental details such as systematic measurements of gate voltage dependence and the dispersion relation that would allow us to explain why ANE is much larger than AHE in our samples.

Comment: Reviewer #3 (Remarks to the Author):

The authors manuscript focuses on nonreciprocal responses (with respect to the magnetization orientation) in Van der Waals heterostructures. Magnetization is induced in the sample by bringing a few-layer insulating ferromagnet (Cr₂Ge₂Te₆) close to the heterostructure (WTe₂). As such, the resulting system may host anomalous Nernst effect (ANE), anomalous Hall effect (AHE), and anisotropic magnetoresistance effect (AMRE), all of which were measured and reported by the authors. The authors then go on to analyze their results distinguishing between bulk and edge responses of anomalous Nernst effect and AHE.

While I find the manuscript interesting, there are significant deficiencies which give me pause in recommending it (see major comments below: e.g., authors claim an AHE signal in Figure 2, but it is unclear from their schematic in Fig. 2a that it is indeed of a "transverse" Hall nature. They seem to drive current between contacts 3-8, and then measure 4-7; while antisymmetrization may pick out signals that are antisymmetric, how is this transverse, how does it show AHE?).

Reply: This is an excellent question. We detected a Hz-antisymmetric hysteresis component in the longitudinal channel (from 4 to 7) and attribute this to the AHE signal mixed in the longitudinal signal. In this regard, this field-antisymmetric component has its transverse origin. It is not completely clear how this signal gets mixed in the longitudinal channel; however, it is conceivable that the non-standard Hall bar geometry is responsible for producing a transverse

component of the current, thus picking up this field-antisymmetric voltage that can only arise from the transverse response, i.e., the AHE signal.

Comment: Another major place for pause is that I do not sufficiently understand what conclusion is new here in this manuscript as compared with what has been done previously, e.g., Ref. 27. In Ref. 27, a magnetic proximity effect was seen in WTe₂ (albeit using CrI₃). What does the present manuscript uncover about physics that is different (it seems that Ref. 27 already had nonreciprocal magnetoresistance). Is it Nernst effect and Hall signals (if so, what new physics does this reveal)? Is it the new magnetic substrate of CrGeTe (if so what new physics does this tell us about, how is this different from CrI₃)? Is it the difference between bulk and edge contributions (if so, what new physics does this teach us?). It would be helpful for the reader to know what is the value proposition of this new paper.

As a result, I do not recommend this paper for publication in its current form. Additionally, while the data is certainly interesting, unfortunately, I found the manuscript a little haphazardly written.

Reply: These are fair criticisms and we hope we have rectified the deficiencies in the revision. The main value proposition of our manuscript is that we unequivocally demonstrated proximity-induced ferromagnetism in the monolayer WTe₂ via a set of well-accepted measurements, i.e., ANE, AHE, and AMRE, resulting in spin-polarized edge state transport which is different from the bulk. This has not been reported in any published work. Here we wish to contrast our work with Ref. 27. First, the observed nonreciprocal response in magnetoresistance reported in Ref. 27 is NOT equivalent to presence of induced ferromagnetism. Examples of nonreciprocal or unidirectional magnetoresistance can be found in heavy metal/ferromagnetic metal (C. O. Avci et al., Nat. Phys. 11, 570 (2015)) and TI/MTI heterostructures (K. Yasuda et al., PRL 117, 127202 (2016); Y. B. Fan et al., Nano Lett. 19, 692–698 (2019)). Second, while the origin of the reported nonreciprocal transport in Ref. 27 was not clear, in our experiments, we know for sure that there is induced ferromagnetism which results in nonlinear and unidirectional transport. We concluded here that ANE can explain the nonlinear (second harmonic) and unidirectional or nonreciprocal transport due to current-induced heating. We went out of our way to establish induced ferromagnetism which might have seemed too messy, but necessary. Additionally, Ref. 27 speculated that ANE plays no role in the nonreciprocal magnetoresistance. Instead, they proposed that electrons in the edge channel interact with magnons in CrI₃, which causes the non-reciprocity observed in their experiments. This effect does not require induced ferromagnetism in WTe₂ either in the edge or bulk. However, it is clear that our ANE, AMRE, and AHE results proved that the entire WTe₂ is magnetized, and they have distinct transport characteristics.

We made changes in the abstract to strengthen the main conclusion of the manuscript.

To avoid distraction in presenting the main findings, we have moved the detailed discussions about SSE vs. ANE and anisotropic magnetoresistance (AMR) vs. spin Hall magnetoresistance (SMR) to SI. As a result, the presentation does seem to be more coherent. We thank the reviewer for making the useful point.

Major comments:

1. In the paper, much attention focussed on AMRE in the heterostructure. However, similar

results are obtained and reported in Ref. 27. What are the main differences between the Author's results and Ref. 27, apart from using CrGeTe in the heterostructure (as opposed to CrI3)? Additionally, I note that Ref. 27 also discussed a possibility of ANE as a mechanism for their non-reciprocity (see e.g., second last paragraph of Ref. 27 main text).

Reply: The discussion on AMRE may seem to be a bit too much to readers outside spintronics community, as the reviewer indicated. It is true that Ref. 27 also showed similar magnetoresistance behavior, but no discussion was placed on the nature of the magnetoresistance. We know that the anisotropic magnetoresistance (AMR) is a property of ferromagnets. From our analysis, we specifically concluded that it is the anisotropic magnetoresistance which must arise from induced ferromagnetism in WTe₂, regardless of what ferromagnetic material induces it. This is in contrast with the spin Hall magnetoresistance effect, which does not require WTe₂ to be magnetized! The origin of the magnetoresistance has been a topic of intense debate in spintronics. We have moved the discussion to SI.

In Ref. 27, the ANE possibility was argued against in the second last paragraph as pointed out by the reviewer. They specifically stated that the transverse temperature gradient by the edge or contacts is unlikely; therefore, they proposed the electron-magnon interaction in the edge channels. It is not hard to see that this mechanism is very different from ours. On this small scale, the temperature distribution is complicated. The sample is actually a heating element. Along the edge of the sample, a lateral temperature gradient is most likely exists as discussed in our manuscript. The correspondence in temperature dependence for both edge and bulk further confirms the similar ANE origin between the nonlocal and self-heating.

2. How do I interpret (any of) the results of Fig. 2 as an AHE? it seems that current was driven between 3-8, and then voltage was then measured between 4-7. Aren't all these contacts longitudinal? What is "Hall" about them? It is not clear that antisymmetrization necessarily captures "Hall" (see top of page 5 of text); if it does, the authors should thoroughly justify.

Reply: This is a good question. Only in ideal Hall bar devices, the longitudinal and Hall voltages can be well separated. Please note that our device is not a regular etched Hall bar (to avoid any etching induced damages); therefore, the longitudinal voltage unavoidably contains transverse voltage due to the irregular WTe₂ shape. When the magnetoresistance is not strictly field-symmetric as it should be, there is a Hall signal mixed in. In the raw data (Fig. 2b, top), it is clear that it contains a Hall hysteresis loop that saturates at two different levels on the positive and negative sides. We know that even for the nonreciprocal transport in edge channels, the 1f-component of AC measurements should be zero (only the second harmonic component is not zero). Hence, this field-antisymmetric contribution in the 1f-signal must come from the Hall effect mixed in the longitudinal channel. To separate the Hall effect, we performed antisymmetrization, which is shown in Fig. 2b. Thank the reviewer for suggesting to explain this procedure. We added a sentence in the first paragraph on page 5 about the need to use Hall bars to have a clear separation of both signals. We must point out that by not etching the sample we cannot get the full Hall voltage magnitude, but only the relative hysteresis changes as the temperature is varied.

3. It would seem that a prominent message of the paper is the claim regarding edge states, last line of main text: "The nonzero edge ANE responses indicate that the ferromagnetic QSHI edge states are qualitatively different from the 1D ballistic chiral or helical edge states." The result seems to be too rapidly generalized and too vaguely discussed. For instance, if one has a quantum Hall system with chiral edge states, wouldn't one expect a Nernst effect? Similarly, wouldn't a QAHE system also display Nernst signals as well? Why is the comparison with 1D even useful?

Reply: We agree that the statement is not very clear and we have rephrased this sentence as "The nonzero edge AHE and ANE responses indicate that the edge states of the magnetized QSHI are partially spin-polarized, qualitatively different from the 1D ballistic chiral edges in QAHEs or helical edges in QSHIs." Our main message is that the edge states in our materials are different from the 1D ballistic edge states. There is a nonzero but less than 100% spin polarization and the edge transport is diffusive. Nevertheless, the edge states are definitely present, which are different from the bulk states. We are aware that in quantum Hall systems, the Nernst effect does exist when the chemical potential sweeps through different Landau levels (H. Nakamura et al., *Solid State Comm.* 135, 510 (2005); Z.W. Zhu et al., *Nature Physics* 6, 26 (2010)). In QAHE systems, the required temperature is very low (< 1 K), which makes the ANE measurements very challenging. In fact, no one has ever succeeded in measuring ANE (theoretically it should be zero), even though the ANE measurements at higher temperatures (above the QAHE phase) have been performed and reported (Minghua Guo et al. *New J. Phys.* 19 113009 (2017)). Our point here was that we have 1D edge states with backscattering that are different from the 1D edge states in QHE or QAHE systems.

Minor comments:

1. The paper has a strange organization. The analysis starts from the discussion of ANE, then shifts to the debate on AHE and AMRE and then returns to the discussion of ANE. This format of narration was harder to follow, including the fact that the results and discussion of FIG. 2 seem to repeat the message of Ref. 27, I would recommend modestly restructuring the paper to make clear what the main point of the paper is.

Reply: The reviewer's points are well taken. To readers who do not work in the spintronics field, the order of the presentation may seem somewhat awkward. Our intention was to first establish the induced ferromagnetism after presenting ANE, AHE, and AMRE data. It may not be fully appreciated that the debate on AHE and AMRE is a vital one to researchers in spintronics. With magnetoresistance, AHE-like and ANE-like data, one question immediately arises: do these effects necessarily mean there is induced ferromagnetism in WTe₂? The answer is not so straightforward without analysis. We actually value these results and discussions because our monolayer WTe₂ devices provided rare and unique samples to address this question effectively. In Ref. 27, the magnetoresistance jumps do look similar to ours, but no discussion was devoted to the nature of the magnetoresistance. For example, they stated on page 504 "the conductance jump when the magnetic state of the CrI₃ changes". They analyzed the jump fields of the CrI₃ trilayer assuming interlayer exchange interaction, which was nevertheless done correctly. It

should be noted that nowhere in this part of the paper did they ask whether the WTe₂ itself behaves ferromagnetically or whether the WTe₂ simply picks up the magnetic state information in CrI₃ via other routes such as spin currents. There are abundant examples in spintronics that the latter case is true, in which the non-magnetic layer simply converts the spin current to charge current or voltage. In our view, this discussion is a unique contribution to the spintronics community. In the revision, we have moved some detailed discussions to SI to avoid distraction.

We should point out that because the induced ferromagnetism in WTe₂ was not a point of Ref. 27 they interpreted the nonreciprocal behavior by proposing electron-magnon interaction across the interface. On the contrary, the non-linear (i.e., 2f-responses) and nonreciprocal responses in our experiments are just a natural consequence of magnetized QSHI, i.e., ANE. In doing so, we have maintained self-consistence.

2. The blue curve plot on FIG. 1f has a strange non-monotonicity at low temperatures. However, this was not commented on in the paper. Clarification of what can be the reason for such behaviour would be beneficial.

Reply: This is a good observation. We thank the reviewer for pointing this out. Fig. 1f includes the lowest temperature (~12 K) data. The raw hysteresis loop data at 12 K actually are noisier than those in Figs. 1d and 1e. We added error bars to the curves in Fig. 1f.

3. FIG 4b is missing a curve that corresponds to the "Bulk+Edge contribution" in the main panel. Is it only in the inset? If it is only in inset, why?

Reply: There are actually three curves. The red curve is completely behind the green ones. We have changed the symbols now so it can be seen in the figure. Based on our analysis in Supplementary Section 10, when the edge conductance dominates ($G_E \gg G_B$), the ANE in Bulk+Edge channel is approximately that in the Edge channel, which is why two curves are almost on top of each other in the plot. Since there are sign changes for the ANE signals in Bulk+ Edge and Edge channels, we cannot plot the y-axis with the log scale to emphasize the small difference, but we zoom in the data in the inset to show the difference at high temperatures.

4. On page 8: authors write "However, a strictly 1D system should not produce any diffusive transverse transport responses such as AHE or ANE signals" What do they mean? It is clear that the system they are studying is not 1D since it is 2D WTe₂ on CrGeTe, so what is the purpose of this statement.

Reply: This could have been made clear. In diffusive 1D channels, there is no width, so there is no transverse response across the 1D channels. For strictly 1D ballistic channels like in quantum Hall or quantum anomalous Hall edges, even though there is no width in the edge channels, they are nonreciprocal and they go around the edge of the sample. This produces the Hall or Nernst signals across the width of the whole sample. It does not happen here because we have diffusive (with backscattering) edges.

On page 8, we rephrased this sentence as “diffusive transport in strictly 1D channel neither produces a response across the channel width for the lack of finite width nor the sample width”.

Reviewers' Comments:

Reviewer #1:

Remarks to the Author:

Li et al have made major revision to their manuscript. The manuscript does read more coherent with a center message. My major criticisms from the last round are adequately addressed in the revision and the rebuttal letter.

The main results now are the induced ferromagnetism in monolayer WTe₂, which are supported by the magnetic hysteresis in ANE and AHE, and the large ANE which is attributed to the edge states.

My remaining criticism are: 1) that the edge part of the ANE is not directly measured but is obtained after subtracting off the bulk contribution; and 2) spin-polarization of the edge states is inferred but not measured. For these two, I think some discussion about possible alternate explanation for the ANE is necessary. 3) The proposed picture of a magnetized quantum spin Hall insulator, which is between the quantum spin Hall and quantum anomalous Hall insulator, still bothers me. It will be great service to me and Nature Communications readers if the authors can provide an illustration of the bandstructure of monolayer WTe₂ after magnetic proximity and how its edge states cause ANE.

I cautiously recommend its publication.

Reviewer #2:

Remarks to the Author:

The authors answered most questions satisfactorily. However, the following issues are unresolved. I understand that points 1 and 2 are inherent deficiencies that cannot be easily addressed with more experiments. I suggest that the authors soften the language to include possible alternative interpretations of their data.

1. Although CGT and WTe₂ are separately a magnetic insulator and a QSHI, both properties are expected to be modified by the proximity of the materials in the heterostructure. There will be charge transfer and even wave function hybridization, which can change the insulating nature of CGT, and QSHI nature of WTe₂. There is really insufficient evidence confirming that WTe₂ on CGT is a magnetized "QSHI".

2. Further, charge transfer could make CGT more conducting, and AHE/ANE could even come from CGT itself. Unlike XMCD which contains spectroscopic information to pinpoint the origin of induced magnetization, transport measurement cannot easily separate the contribution of individual layer in the heterostructure.

3. The AHE/ANE with close to zero remanence and close to zero H_c clearly showed hard axis loops. If easy axis is out of plane (perpendicular anisotropy), the sample should show nearly full remanence with H_c dictated by anisotropy, whether anisotropy is small or large. It should not split into multidomain state at 0 field as it is energetically unfavorable. Even soft magnetic films with much larger thickness would show square hysteresis if it has perpendicular anisotropy.

Reviewer #3:

Remarks to the Author:

The authors have made significant efforts to improve the manuscript. It reads much better now. I also liked their reply which I thought was well written. I particularly liked their reply to my comment about differentiation between their work and Ref. 27 where they try to draw a distinction between measuring non-reciprocal response (in Ref 27) and what they do in their work. For instance, they pointed out the references to Nat Phys 11, 570 (2015) and PRL 117 127202 (2016) and Nanoletters 19, 692-698 (2019). Unfortunately, I couldn't find a discussion of this in their revised text, nor did I find a specific main text citation of the works they brought up in the reply. I felt their explanation was helpful in the reply but was disappointed it didn't appear in the main text.

While I think the manuscript is OK now and actually have no objection to the publication of their manuscript in Nature Communications, there are still some minor statements in their paper that I do not understand — this may be my ignorance so I don't wish to make a big deal about it. However, I raise it below just for the information of authors, in case they find it useful to adjust their phrasing.

For instance, I do not understand why it is so imperative to assert that ANE does not appear for 1D ballistic edge channels. In a QAHI that has chiral edge channels (or even a quantum Hall insulator with the Fermi energy firmly in the gap between Landau levels), don't you expect that it would exhibit a Nernst effect? I can't seem to see why their experiment is any different (for instance see Fig. 1b schematic). I understand the authors use this to claim that this is evidence for a finite width of edge channels for a diffusive system; I'm confused as to why this is important. I'm clearly missing something.

Another place is the logical structure of the penultimate paragraph. The authors say "However, it is not obvious how this mechanism can reconcile with the low temperature $2f$ sign change, as it requires reversal of the non-reciprocity for fixed directions of the temperature gradient and magnetization. Therefore, we conclude that the spin-polarized edge channel of our magnetized QSHI do not behave as the ideal 1D ballistic edges" Why do you need "therefore"? When you break TRS by magnetizing the QSHI, don't you break any and all topological protection from backscattering in the non-magnetized system? Why is this — I.e. the sign changing $2f$ signal — a signature of "non-ballistic"?

Reply to Nat Comm Review Report (NCOMMS-22-09682A)

We thank the three reviewers for taking the time and effort to generate a timely report. We appreciate the encouraging assessments of our manuscript by all reviewers and have made considerable effort in addressing important questions/comments raised in the report. Below please find our point-by-point reply (comments in blue and reply in black). The changes are implemented in the revised manuscript highlighted in the track-change mode.

REVIEWER COMMENTS

Reviewer #1 (Remarks to the Author):

Li et al have made major revision to their manuscript. The manuscript does read more coherent with a center message. My major criticisms from the last round are adequately addressed in the revision and the rebuttal letter.

The main results now are the induced ferromagnetism in monolayer WTe₂, which are supported by the magnetic hysteresis in ANE and AHE, and the large ANE which is attributed to the edge states.

My remaining criticism are: 1) that the edge part of the ANE is not directly measured but is obtained after subtracting off the bulk contribution; and 2) spin-polarization of the edge states is inferred but not measured. For these two, I think some discussion about possible alternate explanation for the ANE is necessary. 3) The proposed picture of a magnetized quantum spin Hall insulator, which is between the quantum spin Hall and quantum anomalous Hall insulator, still bothers me. It will be great service to me and Nature Communications readers if the authors can provide an illustration of the bandstructure of monolayer WTe₂ after magnetic proximity and how its edge states cause ANE.

I cautiously recommend its publication.

Reply. We greatly appreciate reviewer's encouraging assessment of our revised manuscript and the cautious recommendation. We address the remaining questions below.

1) It is true that the edge ANE is not directly measured. It would be ideal to make direct electrical contacts to the edge channels only, similar to the edge contacts in etched graphene devices. However, due to the sensitive nature of WTe₂ and CGT, we deliberately avoided etching by putting the composite layer directly on top of the pre-patterned electrodes. We were concerned that the edge channels are quite narrow (< 100 nm) and it would be literally impossible to place this narrow WTe₂ edge channel right on top of the end of the pre-patterned electrodes. On the other hand, we could place the Bulk electrodes well beyond the edge channel by insulating from the edge channel. In this way, we have pure Bulk electrodes and mixed Edge + Bulk electrodes that allow us to separate them. Although this is not the ideal solution; however, we were able to separate the two contributions using both sets of electrodes, especially at low temperatures when the bulk resistivity is much larger than the edge+ bulk resistivity.

2) The reviewer is right about the spin polarization. Indeed, measuring spin polarization is a quite elaborated job (e.g., via the Meservey-Tedrov experiment, the point-contact Andreev reflection experiment, or magnetic tunneling junction experiment) which would take a considerable amount of dedicated effort. We do not attempt to claim the actual degree of spin polarization, but we do know that it is not zero, nor 100%. In QSHI, the net edge spin polarization is zero due to the two counter-propagating helical edge currents; but in QAHE, the only chiral edge channel gives 100% spin polarization and quantized AHE. We have a situation between these two extremes. On page 9, we added the following discussion about the spin polarization “Under broken time-reversal symmetry, the counter-propagating helical Dirac edge states in magnetized QSHI do not have the same conductivity. In fact, the unquantized AHE must arise from a net charge current composed of two unequal counter-propagating flows of charge carriers, or a spin-polarized current.”

About the alternative explanation of the ANE, the reviewer might suggest adding our reply to his/her question in the last report, i.e., the stray field effect vs. the spin-orbit coupling effect. We thank the reviewer for the suggestion. We estimated the effect of the stray field from the CGT flake on the Nernst signal in WTe₂ using the linear background signal as the “sensitivity” curve and included a brief discussion in the last version (on page 4). We have also added a sentence and a cross-reference on page 4 to indicate the similarity to the origin of the anomalous Hall effect.

3) We understand reviewer’s interest in seeing an illustration of the band structure. The actual band structure shown in the SI (Fig. S7) is calculated for 1T’-WTe₂ in proximity with bulk CGT without including edge states, and Fig. S8 is the band structure including edge states. The edge state dispersion opens a small gap at the Dirac point. The proximity effect also produces asymmetry in the edge dispersion. At the request of the reviewer, we have made an illustration figure (inserted here as Figure R-1) and inserted it as Fig. S9 in the Suppl. Info.

(i) We use the following schematics to illustrate the band structure change on both edges. In edge band structures, the k to $-k$ asymmetry is caused by both the breaking of inversion symmetry and time-reversal symmetry. Because two edges break the inversion symmetry in an opposite way, Edge 2 exhibits larger (smaller) Fermi velocities (v_F) in the right (left) movers than Edge 1, as shown in Figs.S9 b-c. In the scattering picture, Edge 2 behaves more (less) conductive due to longer (shorter) mean free path for the right (left)-flowing current than Edge 1. Therefore, a Hall voltage appears with $V_H > 0$ ($V_H < 0$, $V_H \equiv V_{Edge2} - V_{Edge1}$) for the right (left)-flowing current, presenting the edge-induced AHE.

(ii) We note that only the Dirac bands cannot generate ANE. The Mott’s relation, $\frac{\alpha_{xy}^z}{T} \Big|_{T \rightarrow 0} = -\frac{\pi^2 k_B^2}{3|e|} \frac{d\sigma_{xy}^z}{d\mu}$, requires that the σ_{xy}^z is chemical potential (μ) dependent. To generate both asymmetry and energy-dependence in V_F , the Dirac bands should hybridize with ordinary edge states and/or bulk sub-bands at the edge, to which we refer as quasi-1D states. For example, Fig. S9b demonstrates the strong V_F asymmetry and energy dependence among both Dirac bands and ordinary edge states in the hybridization region. In addition, the energy-dependence of AHE can be further enhanced by the channel number, i.e., density of states.

In summary, the magnetic proximity leads to V_F asymmetry and energy dependence in the quasi-1D edge state (Dirac and ordinary). The V_F asymmetry leads to AHE and the energy dependence of AHE leads to ANE.

Figure R1. Schematics of edge states. **a**, Illustration of band structure of two-dimensional topological insulator (2D TI). **b**, **c**, Band structures of the 2D TI at Edge 1, the lower edge (b), and Edge 2, the upper edge (c), of the device shown in (d) and (e) after the magnetic proximity effect is introduced. Blue and red dashed curves represent the Dirac and the ordinary edge states, respectively, and the light-yellow regions are the bulk states. Consequently, the edge dispersions exhibit asymmetry between Edge 1 (b) and Edge 2 (c). **d**, Edge conductance asymmetry between Edge 1 and Edge 2 due to the Fermi velocity asymmetry. For the Fermi level at position μ_1 , Edge 1 is more (less) conductive than Edge 2 for the right (left)-flowing current, leading to a different voltage (V_H) between the two edges. The thickness of arrows represents the magnitude of edge current flow. **e**, Edge conductance asymmetry for the Fermi level at position μ_2 . Due to the presence of the ordinary edge states (red), there is a competition between conductance at Edge 1 and Edge 2. It is possible to have a sign change for the transverse voltage signal.

Reviewer #2 (Remarks to the Author):

The authors answered most questions satisfactorily. However, the following issues are unresolved. I understand that points 1 and 2 are inherent deficiencies that cannot be easily addressed with more experiments. I suggest that the authors soften the language to include possible alternative interpretations of their data.

1. Although CGT and WTe₂ are separately a magnetic insulator and a QSHI, both properties are expected to be modified by the proximity of the materials in the heterostructure. There will

be charge transfer and even wave function hybridization, which can change the insulating nature of CGT, and QSHI nature of WTe₂. There is really insufficient evidence confirming that WTe₂ on CGT is a magnetized “QSHI”.

Reply. We thank the reviewer for the overall positive assessment. Below we will address these questions one by one.

Charge transfer could in principle take place at the interface but indeed it is very difficult to quantify the effect as the reviewer indicated. However, we respectfully disagree with the reviewer about the effect of the charge transfer on CGT conductivity. To bring this point to readers’ attention, we have added a brief discussion in the manuscript about this possibility. More discussions are in the answers to the next question below.

2. Further, charge transfer could make CGT more conducting, and AHE/ANE could even come from CGT itself. Unlike XMCD which contains spectroscopic information to pinpoint the origin of induced magnetization, transport measurement cannot easily separate the contribution of individual layer in the heterostructure.

Reply. Thanks the reviewer for bringing up this important point. We have carefully reexamined our own as well as other published data regarding possible consequences of charge transfer. We believe that charge transfer is unlikely the cause for our observed AHE and ANE signals in WTe₂/CGT. Here are the main reasons supporting this conclusion.

First, let us use a simplistic parallel resistor model, one resistor from WTe₂ and the other from CGT, and each connected with its own voltage source (Hall or Nernst voltage). In Ref. 31, we know that CGT itself has >10 GΩ in resistance below its Curie temperature and the resistance keeps exponentially going up at lower temperatures. If charge transfer makes CGT conductive so that a significant AHE/ANE signal from CGT appears in the measurements, it can only happen when the CGT resistance is comparable with that of WTe₂. This means that charge transfer induced resistance decrease in CGT would have to be by several orders of magnitude!

Now let us look at the effect of the WTe₂/FM interface on the resistance of WTe₂. As previously reported in Ref. 21 by Fei et al. (Nat. Phys. 18, 94 (2017)) and in Ref. 27 by Zhao et al. (Nat. Mater. 19, 503 (2020)), the low temperature resistance of pristine ML-WTe₂ is ~ 50 kΩ. By putting it in contact with CrI₃, a similar ferromagnetic insulator, they observed a significant resistance increase at low temperatures, which was attributed to a gap (~2.5 meV) opened up in the edge states. In our own experiments, we also observed a large resistance (~ 300 kΩ) at low temperatures when ML-WTe₂ is put in contact with CGT (Fig. 3c). The observed resistance increase, rather than decrease, indicates that charge transfer, if any, produces a negligible effect on the electronic state of the CGT surface. To pick up the Hall or Nernst voltage from the affected CGT surface layer itself that overwhelms induced signals in WTe₂, the resistance would have to decrease by at least 4 orders of magnitude. This is not supported by our experimental results.

Second, the bulk and edge channel transport behaves qualitatively differently, especially below 10 K. If charge transfer effect dominates transport properties, we expect the bulk electrodes to

show a more dramatically reduced resistance at low temperatures because of the larger conducting cross-section compared to the edge electrodes. This is in contradiction to what we observed. As shown in Fig. 3c, the bulk electrodes measure $> 100 \text{ M}\Omega$ resistance, in sharp contrast to $\sim 300 \text{ k}\Omega$ resistance in the edge channel. Given the fact that the edge channel is only $< 100 \text{ nm}$ wide, it is unlikely that charge transfer only occurs over this narrow area in CGT.

We thank the reviewer to raise this legitimate question. We are aware of a recent report on effect of charge transfer in graphene when it is brought into contact with CGT (Chau et al., npj Quant Mater. 7, 27 (2022)). Graphene was indeed found to have a larger charge density which was attributed to charge transfer, but no hysteresis was observed. The effect on graphene is apparently very different from that on WTe₂. Even when the charge transfer in graphene was observed, CGT did not show any magnetoresistance or Hall; otherwise it would have been picked up.

To address reviewer's concern, we inserted a discussion on page 3 and another discussion on page 7 to point out that the resistance of ML-WTe₂ increases, rather than the opposite which would be true if CGT becomes conductive, excludes the charge transfer induced conductive surface layer in CGT.

3. The AHE/ANE with close to zero remanence and close to zero H_c clearly showed hard axis loops. If easy axis is out of plane (perpendicular anisotropy), the sample should show nearly full remanence with H_c dictated by anisotropy, whether anisotropy is small or large. It should not split into multidomain state at 0 field as it is energetically unfavorable. Even soft magnetic films with much larger thickness would show square hysteresis if it has perpendicular anisotropy.

Reply. Some of the loops shown in our manuscript indeed appear to be like hard axis loops, but we do not believe they are. As the reviewer pointed out, CGT has perpendicular anisotropy that favors the magnetization to point out of plane. However, the perpendicular anisotropy energy of CGT still has to compete against the dipolar energy which is larger in thicker flakes. Compared with that of FGT, CGT's anisotropy energy ($K_u \sim 5 \times 10^5 \text{ erg/cc}$) is about 50 times smaller at low temperatures. Very crudely speaking, the anisotropy field $2K_u/M_s$ has to be stronger than the demagnetizing field $4\pi M_s$ to keep the single domain state at remanence. For CGT, this is not a clear-cut case. In Fig. S-2 of the Supporting Material of Ref. 31 (shown in Figure R-2 below), the 150 nm CGT flake shows a collapsed hysteresis loop with zero H_c (a, left panel), as opposed to the open loop of the 50 nm CGT flake (b). While this type of closed hysteresis loops is a characteristic of hard axis behaviors of small magnets, but samples with closure domains and vortices also show similar behaviors, as also briefly discussed in Ref. 31.

Figure R-2. Anomalous Hall loops for two CGT samples with different film thicknesses. The data were included in the Supporting Materials of Ref. 31, a paper published by the corresponding author's group. (a) and (b) represent data in 150 nm and 50 nm thick CGT flakes.

It is useful to note that bulk CGT single crystals have a completely collapsed hysteresis loops even when the field is applied along the easy axis direction, as shown in Ref. 30. In the bulk cases, both easy- and hard-axis loops are like straight lines with no sign of open loops. The perpendicular anisotropy can only be identified by looking at the saturation field difference: higher saturation field in the out-of-plane loop than that in the in-plane loop. Hence the thickness trend is consistent with the evolution of the hysteresis loops as the thickness increases. We should also point out that while the shape of the hysteresis loop depends on the sample thickness, our conclusion about the ANE magnitude is unaffected by this variation because it is obtained at saturation.

To address reviewer's point, we added a sentence on page 4 to indicate the loop share variation due to different flake thicknesses.

Reviewer #3 (Remarks to the Author):

The authors have made significant efforts to improve the manuscript. It reads much better now. I also liked their reply which I thought was well written. I particularly liked their reply to my comment about differentiation between their work and Ref. 27 where they try to draw a distinction between measuring non-reciprocal response (in Ref 27) and what they do in their work. For instance, they pointed out the references to Nat Phys 11, 570 (2015) and PRL 117 127202 (2016) and Nanoletters 19, 692-698 (2019). Unfortunately, I couldn't find a discussion of this in their revised text, nor did I find a specific main text citation of the works they brought up in the reply. I felt their explanation was helpful in the reply but was disappointed it didn't appear in the main text.

Reply. We thank the reviewer for the comments on our previous reply to questions related to nonreciprocal voltages or resistances. The main effort was made to sharpen the focus on

induced ferromagnetism and presentation of the consequences, e.g., nonlinear (second harmonic) and unidirectional (or nonreciprocal) transport originating from ANE. In this revision, we took an extra time to ponder on this issue. To address reviewer's comments, we have done major revisions in a couple places.

First, at the end of the introduction (page 3) where we introduce Ref. 27, now we explicitly state "gave rise to interesting questions such as... and the nature of the non-reciprocity".

Second, on page 6, we indicated that our interpretation of the nonreciprocal $2f$ response may be the origin of the observed nonreciprocal nonlinear responses in other studies and added the pertaining references mentioned above as new references 34-36.

While I think the manuscript is OK now and actually have no objection to the publication of their manuscript in Nature Communications, there are still some minor statements in their paper that I do not understand — this may be my ignorance so I don't wish to make a big deal about it. However, I raise it below just for the information of authors, in case they find it useful to adjust their phrasing.

For instance, I do not understand why it is so imperative to assert that ANE does not appear for 1D ballistic edge channels. In a QAHI that has chiral edge channels (or even a quantum Hall insulator with the Fermi energy firmly in the gap between Landau levels), don't you expect that it would exhibit a Nernst effect? I can't seem to see why their experiment is any different (for instance see Fig. 1b schematic). I understand the authors use this to claim that this is evidence for a finite width of edge channels for diffusive system; I'm confused as to why this is important. I'm clearly missing something.

Reply. We thank the reviewer for bringing up the questions regarding the ANE in QAHI and quantum Hall (QH) systems. The reviewer is correct on why we stated that way. The main purpose is to set up an expectation for such idealistic systems simply based on the Mott relation. Since the Hall (or anomalous Hall) conductivity is just a constant in both QH and QAHI systems, the energy derivative of the Hall conductivity would be zero. In any classical systems, the anomalous Hall conductivity depends on energy, so the ANE is not zero. Our system does not show the quantized Hall conductivity, which already differs from the standard QH and QAHI systems. Moreover, the two-terminal conductance is not quantized either, which indicates diffusive edge channels.

As requested by the first reviewer, we have added a new figure in SI as Fig. S9, as discussed in the earlier part of this reply, to illustrate the quasi-1D edge states. We also rewrote the discussion of the edge ANE on page 9. The main idea behind the quasi-1D, not idealistic 1D edge states is the following. From the well-defined ANE signals, we know the anomalous Hall conductivity must have energy dependence. The energy dependence does not come from the Dirac edge states. Dirac edge-bulk hybridization or Dirac edge-ordinary edge hybridization can make the anomalous Hall conductivity energy dependent, although detailed calculations have not been carried out at this point to study this hybridization effect. In this regard, we refer the edge states as quasi-1D edge states which support spin-polarized transport.

Another place is the logical structure of the penultimate paragraph. The authors say "However, it is not obvious how this mechanism can reconcile with the low temperature $2f$ sign change, as it requires reversal of the non-reciprocity for fixed directions of the temperature gradient and

magnetization. Therefore, we conclude that the spin-polarized edge channel of our magnetized QSHI do not behave as the ideal 1D ballistic edges” Why do you need “therefore”? When you break TRS by magnetizing the QSHI, don’t you break any and all topological protection from backscattering in the non-magnetized system? Why is this — I.e. the sign changing $2f$ signal — a signature of “non-ballistic”?

Reply. We completely agree with the reviewer on this important point. We rewrote the discussion section on page 9, which hopefully addresses this concern.

Reviewers' Comments:

Reviewer #1:

Remarks to the Author:

Li et al have made major revision to their manuscript. The manuscript does read more coherent with a center message. My major criticisms from the last round are adequately addressed in the revision and the rebuttal letter.

The main results now are the induced ferromagnetism in monolayer WTe₂, which are supported by the magnetic hysteresis in ANE and AHE, and the large ANE which is attributed to the edge states.

My remaining criticism are: 1) that the edge part of the ANE is not directly measured but is obtained after subtracting off the bulk contribution; and 2) spin-polarization of the edge states is inferred but not measured. For these two, I think some discussion about possible alternate explanation for the ANE is necessary. 3) The proposed picture of a magnetized quantum spin Hall insulator, which is between the quantum spin Hall and quantum anomalous Hall insulator, still bothers me. It will be great service to me and Nature Communications readers if the authors can provide an illustration of the bandstructure of monolayer WTe₂ after magnetic proximity and how its edge states cause ANE.

I cautiously recommend its publication.

Reviewer #2:

Remarks to the Author:

The authors answered most questions satisfactorily. However, the following issues are unresolved. I understand that points 1 and 2 are inherent deficiencies that cannot be easily addressed with more experiments. I suggest that the authors soften the language to include possible alternative interpretations of their data.

1. Although CGT and WTe₂ are separately a magnetic insulator and a QSHI, both properties are expected to be modified by the proximity of the materials in the heterostructure. There will be charge transfer and even wave function hybridization, which can change the insulating nature of CGT, and QSHI nature of WTe₂. There is really insufficient evidence confirming that WTe₂ on CGT is a magnetized "QSHI".

2. Further, charge transfer could make CGT more conducting, and AHE/ANE could even come from CGT itself. Unlike XMCD which contains spectroscopic information to pinpoint the origin of induced magnetization, transport measurement cannot easily separate the contribution of individual layer in the heterostructure.

3. The AHE/ANE with close to zero remanence and close to zero H_c clearly showed hard axis loops. If easy axis is out of plane (perpendicular anisotropy), the sample should show nearly full remanence with H_c dictated by anisotropy, whether anisotropy is small or large. It should not split into multidomain state at 0 field as it is energetically unfavorable. Even soft magnetic films with much larger thickness would show square hysteresis if it has perpendicular anisotropy.

Reviewer #3:

Remarks to the Author:

The authors have made significant efforts to improve the manuscript. It reads much better now. I also liked their reply which I thought was well written. I particularly liked their reply to my comment about differentiation between their work and Ref. 27 where they try to draw a distinction between measuring non-reciprocal response (in Ref 27) and what they do in their work. For instance, they pointed out the references to Nat Phys 11, 570 (2015) and PRL 117 127202 (2016) and Nanoletters 19, 692-698 (2019). Unfortunately, I couldn't find a discussion of this in their revised text, nor did I find a specific main text citation of the works they brought up in the reply. I felt their explanation was helpful in the reply but was disappointed it didn't appear in the main text.

While I think the manuscript is OK now and actually have no objection to the publication of their manuscript in Nature Communications, there are still some minor statements in their paper that I do not understand — this may be my ignorance so I don't wish to make a big deal about it. However, I raise it below just for the information of authors, in case they find it useful to adjust their phrasing.

For instance, I do not understand why it is so imperative to assert that ANE does not appear for 1D ballistic edge channels. In a QAHI that has chiral edge channels (or even a quantum Hall insulator with the Fermi energy firmly in the gap between Landau levels), don't you expect that it would exhibit a Nernst effect? I can't seem to see why their experiment is any different (for instance see Fig. 1b schematic). I understand the authors use this to claim that this is evidence for a finite width of edge channels for a diffusive system; I'm confused as to why this is important. I'm clearly missing something.

Another place is the logical structure of the penultimate paragraph. The authors say "However, it is not obvious how this mechanism can reconcile with the low temperature $2f$ sign change, as it requires reversal of the non-reciprocity for fixed directions of the temperature gradient and magnetization. Therefore, we conclude that the spin-polarized edge channel of our magnetized QSHI do not behave as the ideal 1D ballistic edges" Why do you need "therefore"? When you break TRS by magnetizing the QSHI, don't you break any and all topological protection from backscattering in the non-magnetized system? Why is this — I.e. the sign changing $2f$ signal — a signature of "non-ballistic"?